# A unique multi-synaptic mechanism involving acetylcholine and GABA regulates dopamine release in the nucleus accumbens through early adolescence in male rats

**Melody C Iacino[1], Taylor A Stowe[1], Elizabeth G Pitts[1], Lacey L Sexton[1], Shannon L Macauley[1,2], Mark J Ferris[1]\***

[1]Department of Translational Neuroscience, Wake Forest University School of Medicine, Winston-Salem, United States; [2]Department of Physiology, University of Kentucky College of Medicine, Lexington, United States

**\*For correspondence:**
mferris@wakehealth.edu

**Competing interest:** The authors declare that no competing interests exist.

**Abstract** Adolescence is characterized by changes in reward-related behaviors, social behaviors, and decision-making. These behavioral changes are necessary for the transition into adulthood, but they also increase vulnerability to the development of a range of psychiatric disorders. Major reorganization of the dopamine system during adolescence is thought to underlie, in part, the associated behavioral changes and increased vulnerability. Here, we utilized fast scan cyclic voltammetry and microdialysis to examine differences in dopamine release as well as mechanisms that underlie differential dopamine signaling in the nucleus accumbens (NAc) core of adolescent (P28-35) and adult (P70-90) male rats. We show baseline differences between adult and adolescent-stimulated dopamine release in male rats, as well as opposite effects of the α6 nicotinic acetylcholine receptor (nAChR) on modulating dopamine release. The α6-selective blocker, α-conotoxin, increased dopamine release in early adolescent rats, but decreased dopamine release in rats beginning in middle adolescence and extending through adulthood. Strikingly, blockade of GABA$_A$ and GABA$_B$ receptors revealed that this α6-mediated increase in adolescent dopamine release requires NAc GABA signaling to occur. We confirm the role of α6 nAChRs and GABA in mediating this effect in vivo using microdialysis. Results herein suggest a multisynaptic mechanism potentially unique to the period of development that includes early adolescence, involving acetylcholine acting at α6-containing nAChRs to drive inhibitory GABA tone on dopamine release.

## Editor's evaluation

This is an important demonstration of how α-6 nicotinic ACh receptors regulate dopamine release via GABAergic mechanisms using ex vivo voltammetry recordings coupled with pharmacological manipulations. There is solid evidence provided to show that GABA tends to suppress dopamine release in adolescents but does not affect dopamine release in adults, a finding that is novel and interesting. Together the data will be of broad interest for further understanding multi-synaptic regulation of dopamine release.

**eLife digest** During adolescence, chemicals and cells in the brain undergo significant reorganization. These changes are thought to be why teenagers are often more vulnerable to developing drug addictions and psychiatric disorders. However, it is not fully understood how the brain transforms during this transitional period.

Most of this reorganization takes place in the dopamine system which is responsible for triggering pleasurable sensations, including the feeling of reward after taking drugs. In 2020, a group of researchers found that adolescent male rats released less of the chemical dopamine in a part of the brain involved in the reward pathway than adult rats. But it was unclear what was causing this age-related effect.

To investigate, Iacino et al. – including some of the researchers involved in the 2020 study –blocked a family of receptors called nAChRs (short for nicotinic acetylcholine receptors) in the brain cells of male rats. These receptors bind to a neurotransmitter called acetylcholine which stimulates cells to release dopamine. Iacino et al. found that inhibiting a specific type of nAChR led to a decrease in dopamine in adult rats, but an increase in early adolescent rats. However, this effect was not observed when other types of nAChRs were inhibited.

Iacino et al. found that the adolescent male rats also had higher levels of another neurotransmitter called GABA which blocks the release of dopamine. This led them to hypothesize that the reduced levels of dopamine in early adolescence may be due to increased levels of GABA, which is secreted by specialized cells which also have nAChRs on their surface.

To investigate, Iacino et al. blocked two receptors for GABA that are found on dopamine-releasing neurons before exposing the rats to the nAChR inhibitor. This caused the adolescent rats to release less dopamine following nAChR inhibition, similar to the levels observed in adult rats. These findings suggest that the nAChR inhibitor leads to a rise in dopamine by stopping cells from releasing GABA – but only in adolescent rats.

The work of Iacino et al. demonstrates how the dopamine system differs in adolescence, which may provide new insights in to why teenagers are often more susceptible to addiction. For instance, nicotine, the addictive substance in cigarettes, can also bind to nAChRs and make them less sensitive to acetylcholine. This may reduce the release of GABA, resulting in more dopamine being released which is then sensed as a reward by the teenage brain. However, more research is needed to fully understand how this brain circuit is modulated by nicotine intake.

## Introduction

Adolescence is the transitional period between childhood and adulthood characterized by changes in behaviors, such as increases in risk-taking, sensation seeking, and peer-focused sociality (*Spear, 2000*; *Spear, 2011*; *Spear, 2013*; *Nelson et al., 2005*). These behavioral changes are thought to be evolutionarily advantageous, as they allow for the exploration necessary for a transition to independence (*Spear, 2000*). However, adolescence is also a time of increased vulnerability to the development of a range of psychiatric disorders, including substance use disorders (SUD), schizophrenia, and affective disorders, that can persist into adulthood (*Casey et al., 2008*; *Fareri et al., 2008*; *Steinberg, 2008*). Adolescent onset is also associated with increased severity of a range of psychiatric disorders (*Chambers et al., 2003*; *Kyriakopoulos and Frangou, 2007*; *Andersen and Teicher, 2008*). This may be, in part, because the brain is still developing during this vulnerable period and disturbances, such as drug-taking or stress, may disrupt the natural developmental trajectory. Thus, understanding how the adolescent brain develops in a normative state may help us pinpoint how and why this system is perturbed in disease.

Reorganization of the dopamine system is hypothesized to underlie many of the changes in adolescent behavior, particularly in reward-related learning and decision-making, and the increased vulnerability to developing psychiatric disorders (*Spear, 2000*; *Spear, 2013*; *Nelson et al., 2005*; *Wahlstrom et al., 2010a*; *Wahlstrom et al., 2010b*). Preclinical research indicates that the dopamine system is going through major changes in adolescence. For example, the density of D1 and D2 dopamine receptors in the striatum, firing rate of dopamine neurons, and dopamine synthesis all peak during adolescence (*Teicher et al., 1995*; *Tarazi et al., 1999*; *Philpot et al., 2009*; *McCutcheon et al.,*

*2012*). While reports examining extracellular levels of dopamine in the striatum show both lower and higher levels of dopamine during adolescence (*Badanich et al., 2006*; *Cao et al., 2007*; *Philpot et al., 2009*), sub-second stimulated dopamine release is consistently found to be decreased in the striatum of adolescent rats (*Stamford, 1989*; *Palm and Nylander, 2014*; *Pitts et al., 2020*). Our lab has previously shown that the maximum concentration of evoked dopamine release is decreased in the nucleus accumbens (NAc) core of early adolescent male rats (*Pitts et al., 2020*), though the mechanism responsible for differentially regulating dopamine release in adolescent and adult rats remains unclear.

Since previous research from our lab revealed that the NAc core has the largest difference in stimulated dopamine release between adult and adolescent male rats (*Pitts et al., 2020*), we focused this report on understanding the mechanisms mediating dopamine release in that region. This is of particular interest because the NAc core is essential for regulating reward-related learning and goal-directed decision- making (*Di Chiara, 2002*; *Saddoris et al., 2013*), which are altered during adolescence. Moreover, terminal dopamine release in the striatum does not always correspond with action potentials from ventral tegmental area and substantia nigra projection neurons (*Trulson, 1985*; *Mohebi et al., 2019*). Release is also regulated by local circuitry via both homosynaptic and heterosynaptic mechanisms (see *Nolan et al., 2020*). Functionality of dopamine transporters and release probability (homosynaptic regulation) can both mediate stimulated dopamine release (see *Ferris et al., 2013*). Heterosynaptic mechanisms include neurotransmitter release from local interneurons and projection neurons, such as acetylcholine interneurons, which play a critical role in regulating local terminal dopamine release (see *Cachope and Cheer, 2014*; *Collins and Saunders, 2020*; *Nolan et al., 2020*). For instance, optogenetic stimulation of acetylcholine release drives terminal dopamine release independent of action potential through activation of nAChRs on dopamine terminals (*Threlfell et al., 2012*; *Cachope et al., 2012*). This local control over terminal dopamine release has important implications for how dopamine encodes salient information and drives behavior, and one or more of the regulatory mechanisms that mediate terminal dopamine release may change between adolescence and adulthood.

Extensive evidence indicates sex differences exist in the dopaminergic system and its local regulation (*Kuhn, 2015*; *Becker, 2016*; *Zachry et al., 2021*). Previously, we have demonstrated that female adolescent rats have significantly higher evoked dopamine release in the NAc core as well as other subregions compared to that of adolescent male rats and adult females (*Pitts et al., 2020*). Thus, the relationship between adolescent and adult dopamine release is opposite in females, and, therefore, either (1) the mechanisms explored here will likely not be the same for females as males, or (2) there is a secondary effect that masks this mechanism in females by altering the relationship we observe at baseline. Since these mechanisms are more than likely sexually dimorphic, we focus here on parsing out the mechanism involved in the changes seen within the adolescent and adult male dopaminergic system. Further research, some of which we are conducting, is needed to understand the relationship between male and female rats.

Given the role of local modulators in the dopamine system, variations in heterosynaptic regulation of striatal dopamine release may be responsible for the neurobiological changes seen between adolescence and adulthood. Therefore, the purpose of this study is to investigate the role of local modulators on presynaptic NAc dopamine release of adult and adolescent male rats is the mechanism responsible for driving such developmental changes. Here, we use ex vivo fast scan cyclic voltammetry (FSCV) to compare stimulated dopamine release and local circuitry regulation in the NAc core of adult and adolescent male rats. We then use microdialysis to determine the influence of other extrinsic factors in vivo in mediating developmental changes in the dopaminergic system. Enhancing our understanding of the transitional changes in dopamine regulation may shed light on the mechanisms driving changes in reward-related behaviors, and related increase in vulnerability, that are characteristic of adolescence.

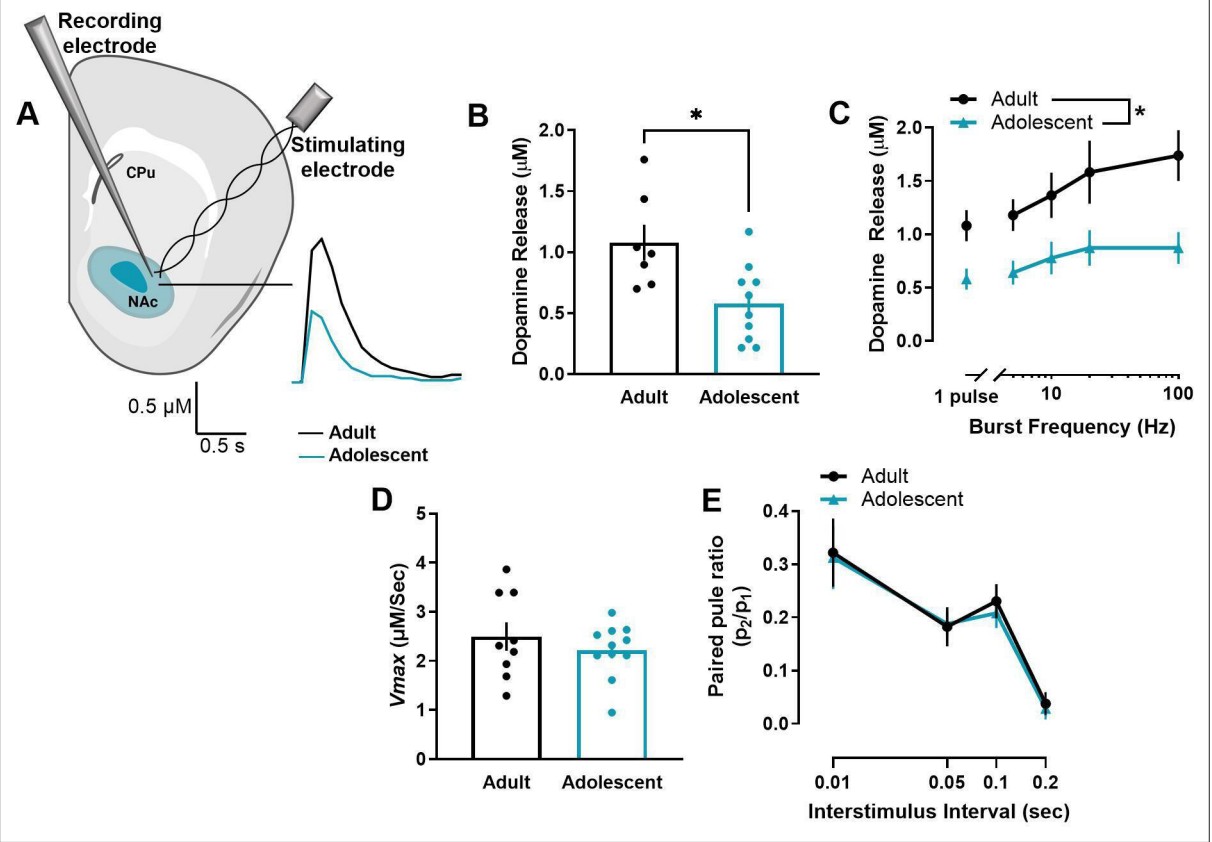

**Figure 1.** Early adolescent male rats have decreased stimulated dopamine release in the nucleus accumbens core. (**A**) Ex vivo fast scan cyclic voltammetry was used to compare the maximum concentration of evoked dopamine release in the nucleus accumbens core of adult (>P70) and early adolescent (P28-35) male rats. Representative traces of single-pulse dopamine release in the nucleus accumbens core (NAc core) are shown. (**B**) Maximum concentration of evoked dopamine release was significantly lower in adolescent than adult rats in the NAc core at single-pulse stimulations and (**C**) across a range of stimulation parameters that model tonic and phasic firing of dopamine neurons (adults: n=7; adolescents: n=10). (**D**) Dopamine uptake rates can impact stimulated dopamine release levels (*Ferris et al., 2013*), so maximal rate of dopamine uptake (*Vmax*) was examined in the NAc core. However, *Vmax* did not differ between adult and adolescent rats (adults: n=9; adolescents: n=11). (**E**) Differences in dopamine release probability can also impact the stimulated concentration of dopamine release (*Ferris et al., 2013*), so we also used paired-pulse ratios to examine differences in release probability. Paired-pulse ratios did not differ between adult and adolescent rats (adults: n=9; adolescents: n=12). In line graphs, symbols represent means ± SEMs. In bar graphs, bars represent means and symbols represent individual data points. Individual data points (n) indicate the number of rats. *p<0.05.

## Results

### Adolescent male rats have decreased dopamine release in the NAc core

We first compared the maximum concentration of stimulated dopamine release in the NAc core of adult (>P70) and early adolescent (P28-35) male rats using ex vivo FSCV (*Figure 1A*). Adolescent male rats have significantly lower evoked dopamine release to single-pulse stimulations ($t_{15}$=2.938, p=0.010) (*Figure 1B*) and across a range of stimulation parameters that model tonic- (5–10 Hz) and phasic-like (20–100 Hz) firing of dopamine neurons (main effect of age: $F_{1, 20}$=10.96, p=0.004) (*Figure 1C*). Differences in dynamics at the dopamine terminal can modulate stimulated dopamine release (see *Ferris et al., 2013*; *Nolan et al., 2020*). For example, activity of dopamine transporters (DAT) have been shown to impact stimulated dopamine levels (*Condon et al., 2019*). To determine whether differences in DAT activity may mediate the age-related differences in stimulated dopamine release, we examined *Vmax* in the NAc core. There was no significant difference in *Vmax* between adult and adolescent male rats ($t_{18}$=0.863, p=0.399) (*Figure 1D*). Probability of dopamine release is another dopamine dynamic that can impact stimulated dopamine levels (*Ferris et al., 2013*). We thus compared paired-pulse ratios, a measure of dopamine release probability (*Cragg, 2003*), between

adult and adolescent rats. However, while interstimulus interval impacted paired-pulse ratios, there was no age-related difference (main effect of interstimulus interval: $F1.84, 34.97=28.64$, $p<0.001$ *Figure 1E*).

## nAChRs differentially modulate dopamine release in the NAc core of adolescent male rats

Since differences in dopamine dynamics at the dopamine terminal such as uptake or release probability did not appear to mediate the age-related differences in stimulated dopamine release, we then hypothesized that local circuitry modulators of dopamine release may be driving the differences between adult and adolescent rats. Acetylcholine, signaling through multiple sub-types of nAChRs, is an important modulator of evoked dopamine release in the NAc core (*Figure 2A*). For example, acetylcholine can drive dopamine release independent of action potentials (*Threlfell et al., 2012*; *Cachope et al., 2012*) and tonically released acetylcholine increases baseline dopamine release (*Zhou et al., 2001*). Additionally, antagonism or desensitization of nAChRs alters dopamine release in a frequency-dependent manner (*Rice and Cragg, 2004*). Given the importance of nAChRs as a local mediator of dopamine release, we applied various drugs that selectively targeted each nAChR sub-type in the NAc core (*Livingstone and Wonnacott, 2009*) and compared changes in the maximum concentration of dopamine released in adults and adolescents. Methyllycaconitine (MLA), a selective α7 nAChR antagonist, impacted dopamine release [expressed as a percentage of pre-MLA (drug-free), single-pulse release] in a frequency-dependent manner, but not in an age-dependent manner (MLA X frequency interaction: $F4, 109.654=4.347$, $p=0.003$) (*Figure 2B, H*). In contrast, α- conotoxin PIA (α-Ctx), a selective antagonist of α6-containing nAChRs, differentially impacted dopamine release (normalized to single-pulse, pre-α-Ctx release) in adult and adolescent rats (age X α-Ctx interaction: $F1, 161.896=27.218$, $p<0.001$), decreasing dopamine release to tonic-like stimulations in adults (1 pulse: $t6=5.743$, $p=0.0012$; 5 pulse 5 Hz: $t6=6.917$, $p<0.001$; 5 pulse 10 Hz: $t6=4.927$, $p=0.0026$) and increasing dopamine release to phasic-like stimulations in adolescents (5 pulse 20 Hz: $t10=2.335$, $p=0.0417$; 5 pulse 100 Hz: $t10=3.276$, $p=0.0084$) (*Figure 2C, H*). When Dihydro-β-erythroidine hydrobromide (DhβE), a β2-selective antagonist, was applied following α-Ctx, it further decreased dopamine release, but did not impact dopamine release differently by age group when plotted as a percent of pre-α-Ctx/drug-free baseline (main effect of DHβE: $F1, 161.022=11.787$, $p=0.001$) (*Figure 2D, H*). Moreover, *Figure 2E* shows that DHβE also has no effect when analyzed as a percentage of dopamine levels that are present following α-Ctx on the slice (i.e. replotted as α-Ctx baseline). To further determine whether α- Ctx is differentially impacting other terminal dopamine dynamics in an age-specific manner, we then examined the rate of dopamine uptake via the dopamine transporter (*V*max) and paired-pulse ratios. α-Ctx did not impact *V*max differently in adult and early adolescent rats ($t17=0.131$, $p=0.898$) (*Figure 2F*). Consistent with extant literature, α-Ctx increased paired-pulse ratios with the greatest increase at the shortest interstimulus interval (α-Ctx X interstimulus interval interaction: $F3,119.685=2.978$, $p=0.034$). However, this effect was not different between ages (age X α-Ctx interaction: $F1, 120.561=0.047$, $p=0.829$) (*Figure 2G*).

We next sought to determine whether age-related differences extend to other periods within adolescence. When we compared dopamine release and the impact of α-Ctx on dopamine release in mid-adolescent rats (P39-42) and adults (P70-90), there was no age-related difference in raw dopamine release (main effects of age and age X frequency interaction: $F<1$) (*Figure 3A, C*) or in α-Ctx modulation of dopamine release (expressed as a percentage of pre-α-Ctx, single-pulse release) in the NAc core (main effect of drug: $F1, 80.984=64.613$, $p<0001$) (*Figure 3B, C*).

## Differences in modulation of dopamine release by α-Ctx in the adolescent NAc core are mediated by GABA in male rats

Antagonism of α6-containing nAChRs impacted terminal dopamine dynamics in an age-specific manner, causing dopamine from early adolescent rats to increase rather than decrease like in adults (*Exley et al., 2008*; *Wickham et al., 2013*). We hypothesized that this difference was not due to differential effects of nAChRs on dopamine terminals, but rather age-related in that the impact of α-Ctx may be driven through another extrinsic modulator of dopamine release. Bath application of α-Ctx increases dopamine release in male adolescents, indicating that blocking α6 nAChRs leads to disinhibition of dopamine release in adolescents, but not adults. GABA has been shown to inhibit terminal

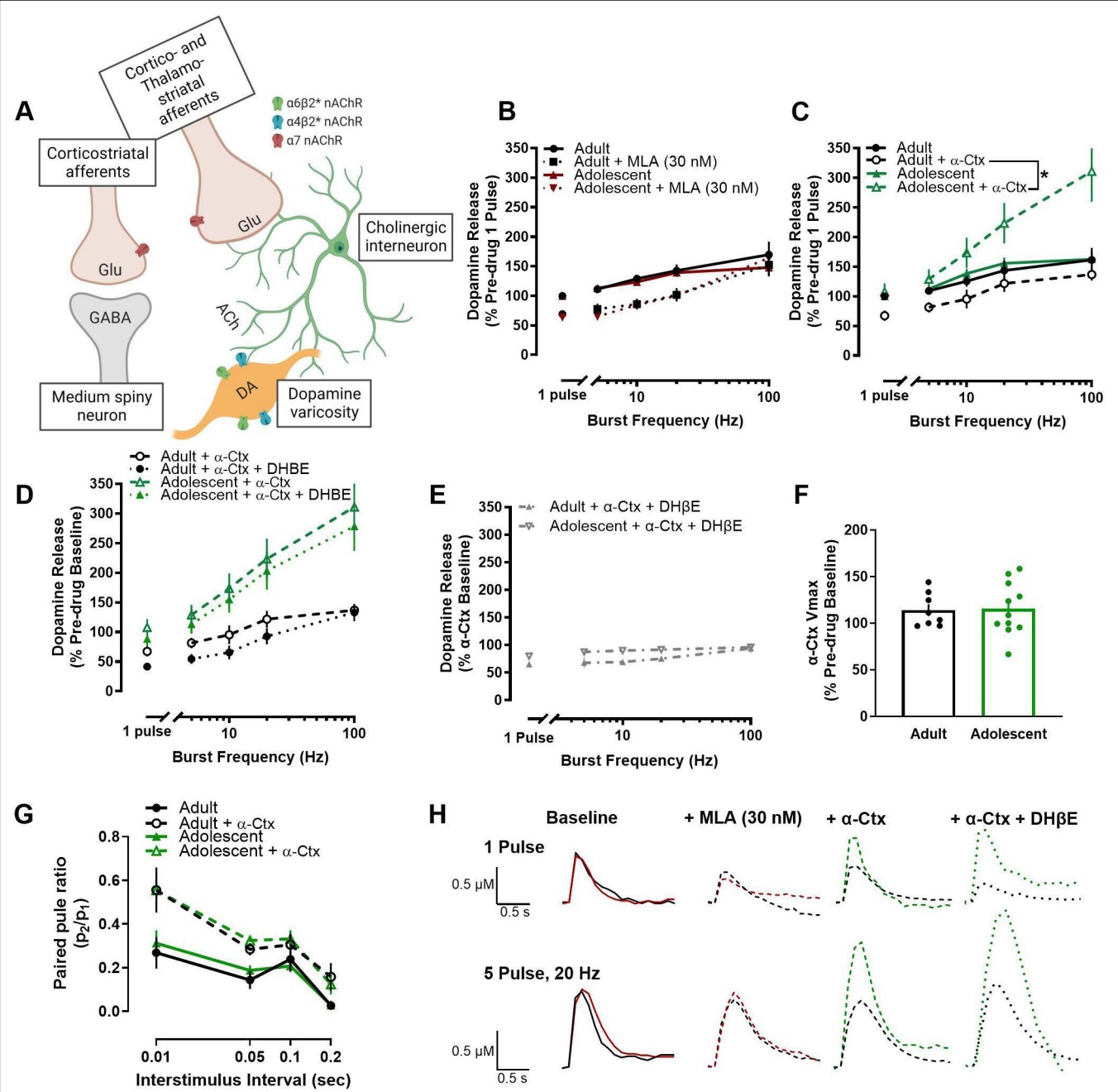

**Figure 2.** Nicotinic acetylcholine receptors (nAChRs), and in particular α6-containing nAChRs, differentially regulate dopamine release in the nucleus accumbens core of adolescent male rats. (**A**) Acetylcholine, signaling through nAChRs located on dopamine terminals, is an important regulator of local dopamine release (***Rice and Cragg, 2004***). There are three main types of nAChRs found in the nucleus accumbens core: α7, α6β2-containing, and α4β2-containing. A schematic shows the localization of the sub- types of nAChRs in the nucleus accumbens core and lists the action of drugs bath applied to brain slices to examine nAChR mediation of dopamine release. We used antagonists specific to various sub-types of nAChRs to examine which nAChRs were playing a role in mediating the age-related difference in nAChR modulation of dopamine release. (**B**) MLA, a selective α7 nAChR antagonist, decreased the maximum concentration of dopamine release, but did not differentially affect the concentration of dopamine release in adult and adolescent rats (adults: n=8; adolescents: n=7). Post-MLA dopamine release is normalized as a percentage of single pulse, pre-drug dopamine release in adults and adolescents, respectively. (**C**) In contrast, α-Ctx, a selective α6-containing nAChR antagonist, facilitated evoked dopamine release in adolescents, but decreased the concentration of dopamine release in adult rats (adults: n=8; adolescents: n=12). Post-α-Ctx dopamine release is normalized as the percentage of single pulse, pre-drug dopamine release in adults and adolescents, respectively. (**D**) Further antagonism of non-α6 β2-containing nAChRs, using DHβE, additionally decreased evoked dopamine release, but not in an age-specific manner (adults: n=8; adolescents: n=12). Adolescent + α-Ctx data is repeated from panel C. Data is presented as a percentage of single pulse dopamine release at baseline for each group to

*Figure 2 continued on next page*

*Figure 2 continued*

assess the combined drug effect. (**E**) Similarly, analyzing DHβE dopamine release as a percent change from the α-Ctx baseline confirmed no age-related differences. (**F**) Given the differential effect of antagonizing α6-containing nAChRs on adolescent and adult dopamine release, we next examined whether α-Ctx changed dopamine dynamics in an age-specific manner. Maximal rate of dopamine uptake (*Vmax*) was not differentially altered by α-Ctx in adult and adolescent rats (adults: n=8; adolescents: n=11). (**G**) Release probability of dopamine, measured by paired-pulse ratio, was increased by α-Ctx, but not in an age-specific manner (adults: n=8; adolescents: n=12). (**H**) Representative traces show dopamine release at single pulse and 5 pulse 20 Hz stimulations in adult (black) and adolescent (red, green) rats. Colors and lines correspond to each drug condition. Symbols represent means ± SEMs. Individual data points (n) indicate the number of rats. *p<0.05.

dopamine release (*Pitman et al., 2014*; *Melchior et al., 2015*; *Brodnik et al., 2019*; *Lopes et al., 2019*) suggesting that α-Ctx could be reducing GABA tone in adolescents. In addition, some striatal GABA interneurons express nAChRs (see *Tepper et al., 2018*; *Figure 4A*; *Figure 5*). To investigate whether the differential modulation of dopamine release by α-Ctx in early adolescent and adult rats is mediated through a multi-synaptic mechanism mediated by GABA, we antagonized GABAA and GABAB receptors, alone or together, prior to the bath application of α-Ctx and then compared the effects of α-Ctx in the presence or absence of GABA receptor antagonism. If the age-related effects of α-Ctx are mediated through GABA receptors, then antagonism of GABA receptors prior to α-Ctx application should eliminate differences in dopamine release modulation between adult and adolescent rats. Application of GABAA and/or GABAB receptor antagonists, alone (*Figure 4B*) or together (*Figure 4C*), did not impact dopamine release and did not modulate dopamine release differently in adult and adolescent rats (main effect of age: $F_{1, 26.121}=0.001$, p=0.981) (*Figure 4B, C*). Importantly, however, application of GABAA or GABAB antagonists, together (*Figure 4*) or alone (*Figure 4E*), prior to the application of α-Ctx completely blocked the differential effect of antagonizing α6-containing nAChRs on dopamine release in adult and adolescent rats (age X frequency X drug interaction: $F_{12, 193.305}=3.988$, p<0.001) (*Figure 4D, E*).

To further test the hypothesis that GABA is mediating the differential effects of α-Ctx in adolescents and adults, we bath applied exogenous GABA prior to the application of α-Ctx. If α-Ctx disinhibits dopamine release in adolescents by reducing endogenous release of GABA, then rescuing GABA levels in the presence of α-Ctx would prevent the age-related effects of α-Ctx in adolescents. Supporting this, we found that GABA prevents the α-Ctx-induced increase in dopamine release in adolescents, causing adolescents to respond to α-Ctx similarly to adult rats (age X frequency X drug interaction: $F_{4, 110.026}=3.189$, p=0.016) (*Figure 4F*). We next analyzed the effects of GABA alone prior to α-Ctx application, although the effects of GABA alone might extend beyond just blocking the α-Ctx effect. We found there were age-related effects of GABA alone on dopamine release (prior

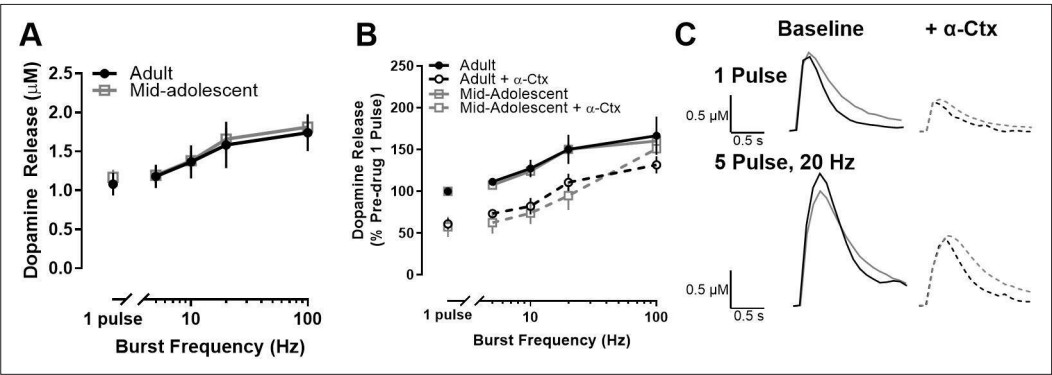

**Figure 3.** Dopamine release and α6-containing nAChR modulation of dopamine release do not differ between mid-adolescence and adulthood in male rats. (**A**) Maximum concentration of stimulated dopamine release in the nucleus accumbens core did not differ between adult (>P70) and mid-adolescent (P39-42) male rats. (**B**) α-Ctx, a selective α6-containing nAChR antagonist, decreased the evoked concentration of dopamine release to a similar degree in adult and mid-adolescent rats (adults: n=7; mid-adolescents: n=4 for A and B). Data are shown as a percentage of pre-α-Ctx, single pulse dopamine release in adults and adolescents, respectively. (**C**) Representative traces show dopamine release at single pulse and 5 pulse 20 Hz stimulations in adult (black) and mid-adolescent (gray) rats with colors and lines corresponding to each drug condition. Symbols represent means ± SEMs. Individual data points (n) indicate the number of rats.

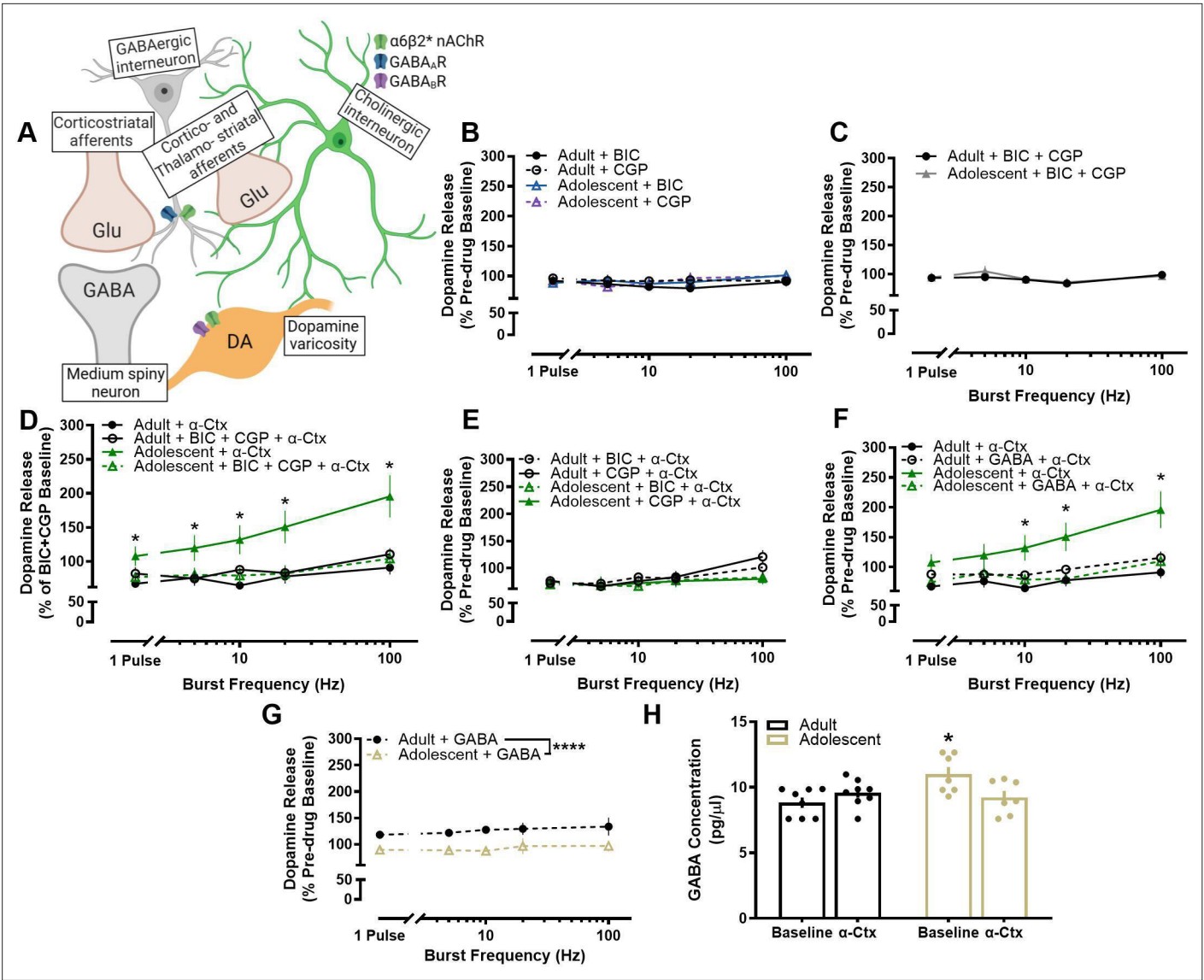

**Figure 4.** Differential modulation of dopamine release by α6-containing nAChRs in adolescent male rats is mediated through GABA receptor signaling. (**A**) A schematic shows localization of nAChRs and GABA receptors in the NAc core. (**B**) Bath application of bicuculline (BIC; a selective GABAA receptor antagonist) and subsequent application of CGP-52432 (CGP; a selective GABAB receptor antagonist) had no effect on stimulated dopamine release (adults: n=6–8; adolescents: n=5–7). Data are shown as a percentage of pre-drug dopamine release to within each stimulation type (i.e. 20 Hz BIC relative to 20 Hz pre-BIC baseline). (**C**) Similarly, analyzing BIC and CGP-52432 application to the pre-drug condition confirmed no effect on stimulated dopamine release (adults: n=6–8; adolescents: n=5–7) (**D**) Prior application of BIC and GCP combined completely blocks the faciliatory effect of α-Ctx on evoked dopamine release in adolescent rats. Effects of each drug on dopamine release are analyzed as a percentage of the pre-drug baseline from their respective stimulation type (e.g. 20 Hz α-Ctx is relative to 20 Hz pre-α-Ctx baseline). Antagonism of GABA receptors did not impact the effect of α-Ctx on dopamine release in adult rats (adults: n=6–8; adolescents: n=5–12). (**E**) Prior application of BIC and GCP following BIC application is not significantly different than the application of both drugs normalized to the pre-drug condition in 4D, indicating both drugs are responsible for blocking α-Ctx effects. (**F**) Prior application of GABA did not impact the effects of α-Ctx in adults, but blocked the α-Ctx-induced increase in adolescent dopamine release (adults: n=6–8; adolescents: n=6–12). Adolescent + α-Ctx data as a percent baseline is repeated from panel D. (**G**) Isolating the effects of GABA on slice prior to α-Ctx application revealed a significant decrease in adolescent dopamine release and an increase in adult dopamine release (adults: 6; adolescents: 5). (**H**) To further support these findings, we then evaluated the effect of infusing α-Ctx into the NAc on GABA neurotransmission using in vivo microdialysis. Adolescents have higher GABA tone as compared to adults (adults: n=8; adolescents: n=8) as indicated by Age X Drug Interaction and post-hoc multiple comparisons. *p<0.05; Sidak posthoc multiple comparisons test. In line graphs, symbols represent means ± SEMs. In bar graphs, bars represent means and symbols represent individual data points. Individual data points (n) indicate the number of rats. *p<0.05, ****p<0.0001.

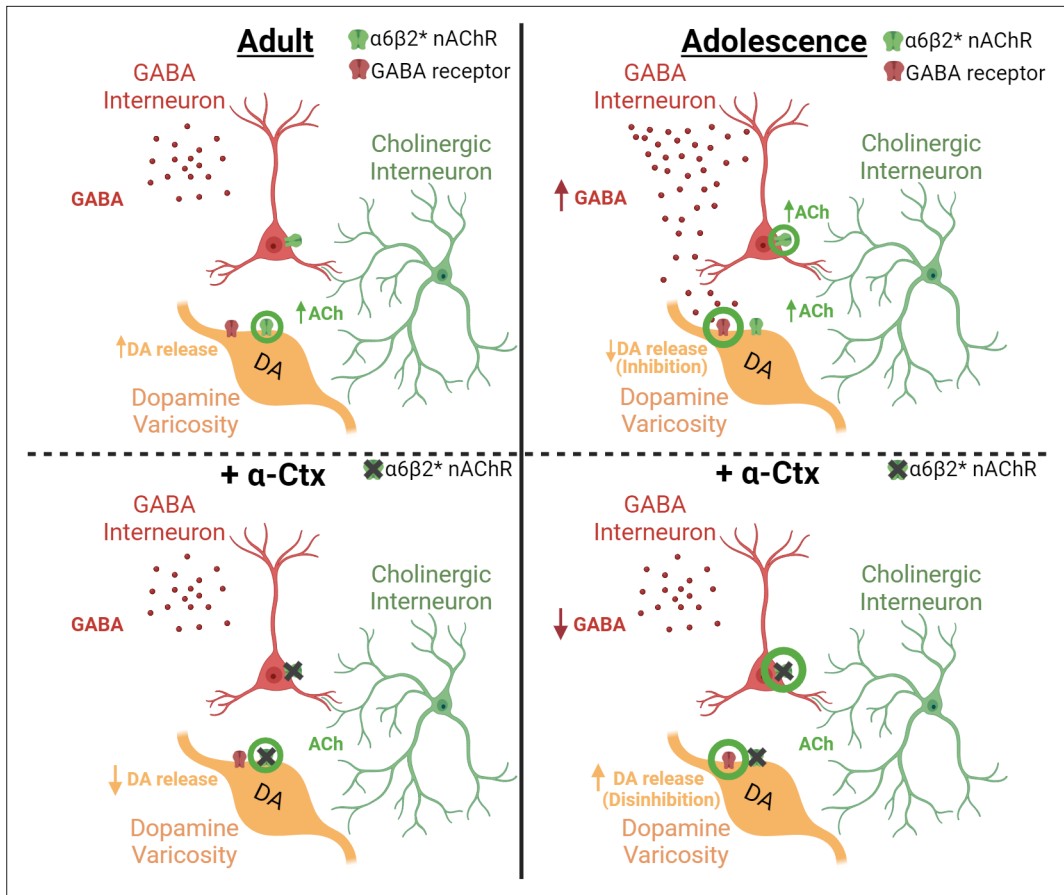

**Figure 5.** Differential effect of α6-containing nAChRs on GABA and dopamine release in the striatum of adult versus adolescent male rats. A schematic shows the relationship between GABA interneurons, cholinergic interneurons (CINs), and dopamine (DA) varicosities in adult (left column) and adolescent (right column) rats before (top row) and after (bottom row) application of α-Ctx. The green circles represents the primary mechanism of dopamine modulation for each panel. The present study demonstrated that adolescents have higher GABA tone than adults (top right vs left panels) suggesting that dopamine release is reduced in adolescent rats via GABA-mediated inhibition of DA varicosities. Applying α-Ctx on slice (bottom panels) blocks α6-containing nAChRs directly on DA varicosities in adult rats (bottom left, green circles), thus preventing direct facilitation by acetylcholine from CINs. In adolescents (bottom right), the blockade of α6-containing nAChRs reduces GABA tone (green circles) and subsequently disinhibits DA varicosities, which masks the direct effect of acetylcholine on DA varicosities observed in adults. This masked mechanism is revealed in adolescent rats when the influence of α6-mediated decreases in GABA is removed via (1) blocking GABA receptors or (2) rescuing GABA levels with exogenously applied GABA.

to α-Ctx), that were opposite that of α-Ctx, albeit with a smaller effect size. GABA decreased dopamine release in adolescent rats but increased release in adult rats (main effect of age: $F1,44=27.58$, $p<0.0001$) (*Figure 4G*).

The effects of GABA on slice alone are not specific to the effects of α6-nAChRs on GABAergic interneurons. Moreover, α-Ctx effects on striatal microcircuits may vary in an intact brain. Therefore, investigating the effects of α-Ctx on GABA tone with in vivo microdialysis allows for the evaluation of age-related differences of extracellular GABA in the interstitial fluid of a live, freely moving rat. If age-related effects of α6- nAChRs are mediated by variations in adolescent GABA tone, then infusing α-Ctx directly into the NAc should demonstrate the α-Ctx-induced reduction of GABA levels. Baseline recordings of GABA levels in the NAc revealed that adolescent GABA tone was significantly higher than adult GABA tone (main effect of age: $F1,27=4.88$, $p=0.0358$) (*Figure 4H*; *Figure 5*). Notably, we found that infusion of α-Ctx into the NAc led to a α-Ctx X Age Interaction ($F_{1,26}=6.408$, $p=0.0178$), but no main effect of α-Ctx ($F_{1,27}=0.69$, $p=0.4109$). Moreover, Sidak's posthoc multiple comparisons

revealed a significant effect between adult vs adolescent baseline GABA (p<0.05), suggesting that α-Ctx modulated GABA tone differently in adults and adolescents (main effect of drug: $F_{1,27}=0.69$, p=0.4109) (*Figure 4H*). These results provide parallel evidence to our finding that the α-Ctx-induced increase in dopamine release in adolescent male rats may be GABA-mediated.

## Discussion

Significant reorganization of the dopamine system occurs during the transitional period between adolescence and adulthood. Adolescents exhibit greater dopamine receptor levels, increased neuronal sensitivity, and altered dopamine dynamics when compared to adults (*Teicher et al., 1995*; *Bolanos et al., 1998*; *Galvan, 2010*). Restructuring of the dopamine system occurs in protein expression and activity that affects dopamine release or uptake. Although these transitional changes in dopamine release are implicated in a wide range of psychiatric disorders, much is unknown about the mechanisms driving these age-related differences. Thus, the goal of this study was to identify variations in NAc core stimulated dopamine release and pinpoint the mechanism for which these changes may occur in adolescent and adult male rats.

Consistent with our previous findings (*Pitts et al., 2020*), we showed no difference in our measure of dopamine uptake (Vmax) between adult and adolescent male rats in the NAc, suggesting the rate of dopamine uptake does not drive differences in stimulated dopamine release shown here. While the affinity of dopamine for the DAT (i.e. Km) has been shown to be different in prior publications by other groups (*Walker and Kuhn, 2008*), the dopamine curves we generated were optimized for measures of release and Vmax, not baseline affinity measurements (i.e. Km). Km is traditionally derived from the bottom third of the descending portion of the dopamine curve in FSCV, and electrode adsorption and other artifacts are known to occlude this outcome measure without affecting our other measures (*Ferris et al., 2013*). Thus, we did not quantify this measure and it remains possible that the affinity of dopamine for the DAT could be different between these groups.

Stimulation parameters were adjusted to model both tonic- and phasic-like neuronal firing across a wide range of frequencies in order to mimic physiologically relevant firing patterns. Replicating our previous findings (*Pitts et al., 2020*), the present study found decreased dopamine release in the NAc core of adolescent male rats. This is consistent with in vivo evidence of male adolescent rats demonstrating lower NAc dopamine release than adults. For example, high-frequency stimulations of the VTA in anesthetized rats reveal significantly greater stimulated NAc dopamine release in adult males compared to adolescent males (*Walker and Kuhn, 2008*). In freely moving rats, differences in dopamine release between early adolescence and adulthood appear stimuli-specific and differ in habituation over repeated exposures (*Robinson et al., 2011*). For example, the frequency of dopamine transients was no different between early adolescent and adult rats both at baseline and in the initial response to social interactions with other rats (*Robinson et al., 2011*). However, the unexpected presentation of additional stimuli across many modalities showed less frequent dopamine responses in early adolescent rats compared to the classic dopamine response of adult rats (*Robinson et al., 2011*). We found that decreases in stimulated dopamine release are not driven by differences in release probability within a burst. For example, the paired-pulse ratio (PPR) shows the relative magnitude of release attributed to each pulse in a single burst. This differs from electrophysiological PPR in that we are directly observing the relative release magnitude from a population of neurons (as opposed to a single cell) between pulse 1 and pulse 2. PPRs are known to be frequency dependent as we show here, but we find there are no age-related differences in the distribution of release magnitude across pulses in a single burst. Of note, we previously found the age-related relationship of dopamine release in the NAc core to be opposite in females compared to males, with higher NAc dopamine release in adolescent females as compared to adult females (*Pitts et al., 2020*). However, research has demonstrated that striatal dopamine release is robustly regulated by ovarian hormone regulation (*Yoest et al., 2018*; see *Zachry et al., 2021*). Given these physiological differences, our proposed mechanism may be masked, or is not representative of the female dopamine system. Thus, we focus the present studies on mechanisms underlying the male adolescent and adult dopamine system with plans to study female mechanisms of dopamine release in future work.

Given no differences in uptake rate or release probability within a burst, we shifted our focus to examining the involvement of heterosynaptic regulators, like cholinergic interneurons (CIN), on local dopamine release. We examined CIN influence on dopamine release under single pulse

conditions and in the aggregate magnitude of release attributed to multiple pulses in a burst that mimics 'tonic-like' or 'phasic-like' firing conditions (i.e. not the relative release across individual pulses in a single burst as with the PPR). Acetylcholine and nAChRs are important regulators of terminal dopamine release within the striatum (see *Threlfell et al., 2012*; *Cachope and Cheer, 2014*; *Collins and Saunders, 2020*). Activation of CINs can cause dopamine release independent of VTA-mediated action potentials (*Threlfell et al., 2012*; *Cachope et al., 2012*), while pauses in CINs amplify the signal-to-background dopamine release by decreasing dopamine to low-frequency activity (*Cragg, 2006*). Furthermore, the antagonism of nAChRs has been shown to regulate dopamine release in a frequency-dependent manner, decreasing dopamine release to tonic-like stimulations, while not impacting or facilitating dopamine release to phasic-like stimulations (*Rice and Cragg, 2004*; *Cragg, 2006*). We found differences between adult and adolescent rats in modulation of NAc core dopamine release by α6- containing nAChRs, but not by α7 or non-α6 β2* nAChRs. Indeed, the application of α-Ctx (a selective α6- containing nAChRs antagonist) had a completely different effect on dopamine release in adults compared to early adolescent male rats. Namely, α-Ctx decreased dopamine release to tonic-like firing in male adults, while increasing dopamine release to stimulation frequencies that model phasic-like firing in male adolescents. A previous study from our lab using adult rats found that those rats most vulnerable to sensation seeking and markers of SUD (*Siciliano et al., 2017*) display a similar pattern of dopamine release to adolescent rats in the present study, particularly with respect to how α6-containing nAChRs modulated dopamine release. This is an intriguing finding and may indicate that α6-containing nAChR regulation of dopamine is a biomarker for vulnerability to SUD given that adolescence is a particularly susceptible time for sensation seeking and SUD development. Additional experiments and studies across more models of vulnerability are necessary to strengthen this comparison.

We next found that antagonists of GABAA and/or GABAB receptors applied prior to administration of α-Ctx blocked the α-Ctx-induced facilitation of dopamine release in adolescents, but not in adults. Blocking GABA receptors in adolescent rats unmasked the α-Ctx mediated decrease in dopamine release at single pulse and lower frequency stimulations that is well documented in prior literature and in the current adult rat group. This suggests that the blockade of α6-containing nAChRs in adolescent rats may have two parallel and independent mechanisms, as illustrated in *Figure 5*. The first is the well-documented direct blocking of α6-containing nAChRs on dopamine varicosities that decreases DA release at low-frequency stimulations (i.e. the 'masked' effect in adolescents but present for adults). The second, and novel mechanism through early adolescence we propose, is an α-Ctx mediated reduction in GABA control of dopamine release that disinhibits dopamine and is responsible for the increase that is specific to adolescent rats. Indeed, nAChRs are expressed on the pre-synaptic terminals of GABA interneurons in the striatum (*Tepper et al., 2018*) and research indicates GABAA (and possibly GABAB) receptors are located on dopamine terminals and can impact both dopamine and GABA co-release (*Figure 5*; *Pitman et al., 2014*; *Melchior et al., 2015*; *Brodnik et al., 2019*; *Lopes et al., 2019*; *Patel et al., 2024*). By applying GABA receptor antagonists prior to α-Ctx, the α-Ctx-mediated changes in dopamine are free from the influence of GABA. This is confirmed by our separate experimental finding that exogenous application of GABA prior to α-Ctx also prevents the age-related differences observed following α-Ctx alone. Thus, we propose that for early adolescent rats (*Figure 5*), α-Ctx blocks α6 nAChRs on GABA releasing cells to lower the influence of GABA on dopamine varicosities upon stimulation. The putative source of this GABA that expresses α6 nAChRs includes both local GABA interneurons and dopamine varicosities that co-release GABA (*Patel et al., 2024*). This leads to disinhibition of dopamine and the elevated levels we observe only in adolescents. This effect masks the parallel mechanism of the direct influence of acetylcholine on dopamine release via varicosities only observed in adult rats. By replacing the reduced GABA levels caused by α-Ctx directly with exogenously applied GABA, or by blocking GABA receptors, we bypass the mechanism of α-Ctx-mediated reduction in GABA and subsequent disinhibition of dopamine. Although prior research has found no difference in the density of α6-containing nAChRs in the NAc of early adolescents and adults (*Doura et al., 2008*), differences in localization or functionality of either α6-containing nAChRs or GABA receptors may underlie changes that create this novel multisynaptic mechanism of dopamine regulation in adolescent males.

Antagonizing GABAA or GABAB receptors, alone or together, did not impact dopamine release directly (i.e. outside of α6 nAChR modulation). In fact, prior literature on the effects of GABA antagonism

on dopamine release is mixed. Some report no direct impact on dopamine release (*Pitman et al., 2014*). However, *Patel et al., 2024* recently confirmed the presence of GABA-mediated inhibition of striatal dopamine release in male and female mice. In that study, the non-competitive antagonist of GABA$_A$R chloride channels (i.e. Picrotoxin) was able to increase/disinhibit dopamine in the dorsal striatum and NAc (*Patel et al., 2024*). Differences in methodology, such as optogenetic versus electrical stimulation of dopamine varicosities, may account for the lack of GABA$_A$R or GABA$_B$R antagonist effect in our study. Given the non-selective nature of electrically evoked dopamine release here, it is possible that additional nearby GABA sources (compared to optogenetic studies) released enough GABA to 'outcompete' the competitive antagonists. Since GABA antagonists were only effective in the presence of α-Ctx, we propose that α-Ctx-mediated decreases in GABA may have lowered GABA enough to unmask the effects of the GABA competitive antagonists. Finally, *Brodnik et al., 2019* found that GABAA receptor regulation of dopamine release in the NAc core functions through GABAB receptor activation. It is possible that a similar mechanism of action is mediating the effects seen here, although we did not investigate varying combinations of activation vs inhibition of GABAA and GABAB receptors. However, given this lack of direct effect of GABA antagonism on dopamine release, we applied GABA on the slice to confirm the role of GABA on the α-Ctx mediated effect as discussed earlier. Overall, activation of GABA receptors using GABA alone on slice led to small, but divergent dopamine release in adolescents and adults. The mechanism for this is not known, but could be differential effects of GABA receptor sensitivity on dopamine varicosities (*Brodnik et al., 2019*) or on CINs (*Yorgason et al., 2022*). Regardless, the small effects of GABA alone on a slice in the presence of endogenous GABA are likely independent of the α6-mediated control of dopamine release explored in the present study.

Lastly, we used microdialysis in awake rats to confirm that the age-dependent differences in male dopamine release are mediated by a GABA mechanism involving α6-containing nAChRs in vivo. Indeed, the effects in vivo were consistent with expectations set by our studies in slices. First, we found that GABA tone is higher at baseline in adolescents compared to adults, which is consistent with greater inhibition of dopamine release in our study (*Figure 5*, top row). Moreover, infusing α-Ctx directly into the NAc altered GABA tone in an age-dependent manner as indicated by the Age X Drug Interaction. Notably, α-Ctx mediated reductions in adolescent GABA levels are comparable to adult baseline GABA levels. Thus, blocking α6 nAChRs in adolescent rats makes GABA and dopamine signaling similar to that of adult rats at baseline in both our microdialysis and FSCV studies. Further studies are needed to understand the microcircuitry mediating the α6-containing nAChR regulation of dopamine release in adolescent rats. Interestingly, one report using adult mice found that antagonizing GABAB receptors increased dopamine release when β2-containing nAChRs were antagonized (*Lopes et al., 2019*), indicating that GABA and nAChRs have a potentially complex and reciprocal regulation of dopamine release that we suggest here may be age/developmental stage dependent.

In conclusion, we found that stimulated dopamine release is decreased in the NAc core of early adolescent male rats. This effect is driven, in part, by multisynaptic regulation of dopamine release only in adolescent male rats, through both acetylcholine acting at α6-containing nAChRs and GABA tone. The dopamine system undergoes reorganization during adolescence (see *Wahlstrom et al., 2010a*; *Wahlstrom et al., 2010b*; *Padmanabhan and Luna, 2014*) and these changes are implicated in behavior as well as increased vulnerability to the development of various psychiatric disorders (*Spear, 2000*; *Spear, 2013*; *Nelson et al., 2005*; *Wahlstrom et al., 2010a*; *Wahlstrom et al., 2010b*). The present study gives us novel insight into age-related differences in dopamine release and its regulation, thus highlighting critical neurochemical targets that may partially underlie these behavioral changes. Examining how these systems develop in healthy individuals is imperative to our understanding of how perturbations may disrupt development and lead to certain disease states.

## Materials and methods
### Animals
Early adolescent (P28-35), mid-adolescent (P38-42), and adult (P70-90) male Sprague-Dawley rats (Envigo, Huntingdon, UK) were maintained on a 12:12 hr reverse light/dark cycle (4:00 a.m. lights off; 4:00 p.m. lights on) with food and water available ad libitum. All animals were ordered in and allowed one week to habituate in the home chamber/housing colony prior to the start of experiments. All

animals were maintained according to the National Institutes of Health guidelines in the Association for Assessment and Accreditation of Laboratory Animal Care accredited facilities (Accreditation #: 00008; PHS Assurance #: D16-00248 (A3391-01)). All experimental protocols were approved by the Institutional Animal Care and Use Committee at Wake Forest School of Medicine (Protocol Approval #: A21-143).

## Slice preparation

Rats were anesthetized with isoflurane and then euthanized by rapid decapitation in a ventilated area free of blood or tissue from previous animals. Brains were rapidly removed and transferred into pre-oxygenated (95% O2/5% CO2) artificial cerebral spinal fluid (aCSF) containing (in mM): NaCl (126), KCl (2.5), monobasic NaH2PO4 (1.2), CaCl2 (2.4), MgCl2 (1.2), NaHCO3 (25), dextrose (D-glucose) (11), and L-ascorbic acid (0.4). Tissue was sectioned into 400 µm-thick coronal slices on a compresstome VF-300 vibrating microtome (Precisionary Instruments Inc, San Jose, CA). Brain slices were transferred to testing chambers containing oxygenated aCSF (32 °C) flowing at 1 mL/min.

## Ex vivo fast scan cyclic voltammetry

Slice FSCV was used to characterize dopamine release in the NAc core (*Fennell et al., 2020*). We used ex vivo FSCV for its many unique advantages, including the precise manipulation of local micro-circuitry. Slice FSCV has high temporal and spatial resolution and can measure the effects of multiple drugs within a single animal (*Ferris et al., 2013*) Briefly, a carbon-fiber recording microelectrode was placed 100–150 mm from a bipolar stimulating electrode. Extracellular dopamine was recorded by applying a triangular waveform (–0.4 to +1.2 to - 0.4 V *vs* Ag/Agcl, 400 V/s), sampling every 100 ms. Dopamine release was initially evoked by a single electrical pulse (750 mA, 2 ms, mono-phasic) applied to the tissue every 3 min. After the extracellular dopamine response was stable (three collections within <10% variability), five-pulse stimulations were applied at varying frequencies (5, 10, 20, or 100 Hz) to model the physiological range of dopamine neuron firing. Additionally, paired-pulse stimulations were used to determine baseline probability of dopamine release. Paired-pulse ratios can be used to determine the release probability of dopamine (Condon et a., 2019; *Cragg, 2003*). We define paired-pulse ratio in a manner consistent with extant literature, namely as the ratio P2/P1, where P1 is peak dopamine release detected following 1 p stimulus and P2 is the peak dopamine release attributable to the second stimulation only. P2 was determined by subtracting dopamine release, including decay phase, after a single pulse from the summed paired-pulse response (i.e. P2-P1/P1).

After assessing the dopamine response to varying stimulation parameters, various compounds targeting nAChRs (Methyllycaconitine [MLA; a selective α7 nAChR antagonist], 30 nM; α-cono-toxin PIA [α-Ctx; a selective α6-containing nAChR antagonist], 100 nM; dihydro-beta-erythroidine [DhβE; a selective β2-containing nAChR antagonist], 500 nM) or GABARs (Bicuculline [BIC; a selective GABAAR antagonist], 10 µM; CGP-52432 [CGP; A selective GABABR antagonist], 5 µM; γ-Aminobutyric acid [GABA], 10 µM) were bath applied and dopamine response equilibrated to single pulse stimulation. Separate slices from the same animal were used to test each drug independently, and the same frequency-response curves assessed under drug-free conditions were reassessed following drug application in each slice. In order to test the distinct contributions of α6* and non-α6* nAChRs, we added α-Ctx and DhβE in a cumulative fashion, equilibrating and testing single and multi-pulse frequencies (described above) following α-Ctx and then DhβE. Changes in dopamine signaling between α-Ctx alone and in combination with DHβE differentiated the contribution of α6* and non-α6* β2-containing nAChRs. Similarly, to examine the role of GABA or GABARs in the effects of α-Ctx, we bath applied GABA or BIC and CGP alone or together. Then, after equilibration and frequency-response curves, we applied α-Ctx and re-tested single- and multi-pulse stimulations. We ran only single doses of each compound because multiple different drugs and/or multiple stimulation parameters were applied on a limited number of slices per animal. Thus, we did not test whether group differences were a result of shifts in potency and/or efficacy. Examining single concentrations and documenting these effects across multiple stimulation parameters and the systems involved in the current study provides impetus to explore these additional outcomes in future studies.

## In vivo microdialysis

Freely moving microdialysis was used to determine how extracellular GABA in the interstitial fluid varies in adults versus adolescents. Microdialysis guide cannulae (BASinc, West Lafayette, IN) were stereotaxically implanted and counter-balanced in a separate cohort of adult and adolescent rats above the NAc. Adult (P70-90) coordinates were anteroposterior +1.2 mm, lateral ±2 mm, and ventral –5.2 mm. Coordinates are relative to bregma, midline, and skull surface, respectively. Adolescent (P28-35) coordinates were calculated individually using a bregma/lambda ratio in comparison with adult bregma/lambda measurements (~9 mm adult distance). For example, if the distance between bregma and lambda was 8.3 mm for a single adolescent, our ratio for that individual would be 8.3 mm/9 mm. The result (i.e. 0.92) would then be multiplied by adult anteroposterior, lateral, and ventral coordinates to calculate new adolescent coordinates. Adolescent coordinates exhibited slight individual variation. All rats were allowed 3–5 days recovery time following surgery before the start of any experimentation. Microdialysis probes (1 mm membrane length; BASinc) were inserted ~1 hr before the first sample collection. Tissue was continuously perfused at 1 µL/min throughout the duration of the experiment with artificial cerebrospinal fluid aCSF (pH 7.4) containing (in mM): NaCl (148), KCl (2.7), CaCl2 (1.2), MgCl2 (0.85). Collections occurred 3 hr into the dark cycle across 6 hr with samples collected every 20 min (i.e. 3 samples per hour). During the first 3 hr, 9 samples were collected as baseline samples. Following baseline collections, α-Ctx (1.6 µL) was then infused at 0.2 µL/min over 8 min. During the 3 hr post α-Ctx, nine additional samples were collected starting 10 min following the start of α-Ctx infusion in 20 min intervals. Upon completion of microdialysis sampling, rats were euthanized and brain extracted. Slices containing the NAc were then used to confirm proper placement of microdialysis probe by recording live tissue probe tracts in all rats.

## General GABA ELISA

General GABA ELISA kit (ABClonal, Woburn, MA) was used to measure extracellular GABA concentration from all sampled microdialysates. All measurements were taken according to the manufacturer's instructions (ABClonal, cat. #RK09124). Baseline samples were assayed in duplicate while post α-Ctx samples were assayed in triplicate for each individual rat. All sample wells were spiked with 0.0003 µg/30 µL GABA while GABA wells were spiked with 0.0005 µg/ 50 µL GABA to make 10 ng/ mL samples to overcome the kit threshold. Samples were frozen at −80 °C until all collections were complete and thawing for ELISA assay.

## Drugs

Methyllycaconitine citrate (20-ethyl-1α,6β,14α,16β-tetramethoxy-4-[[[2-[(3 S)–3-methyl-2,5-dioxo-1-pyrrolidinyl]benzoyl]oxy]methyl]-aconitane-7,8-diol, 2-hydroxy-1,2,3-propanetricarboxylate; Cayman Chemical Company, Ann Arbor, MI), α-conotoxin PIA (Research and Diagnostic Systems, Inc, Minneapolis, MN), Dihydro-β-erythroidine hydrobromide ((2 S,13b*S*)–2-Methoxy-2,3,5,6,8,9,10,13-octahydro-1*H*,12*H*- benzo[*i*]pyrano[3,4 *g*]indolizin-12-one hydrobromide; Tocris Bioscience, Bristol, UK), (-)-Bicuculline methochloride ([*R*-(*R*\*,*S*\*)]–5-(6,8-Dihydro-8-oxofuro[3,4-*e*]–1,3-benzodioxol-6-yl)–5,6,7,8-tetrahydro-6,6-dimethyl-1,3-dioxolo[4,5 *g*]isoquinolinium chloride; Tocris Bioscience, Bristol, UK), CGP-52432 ((3-[[[(3,4- Dichlorophenyl)methyl]amino]propyl] diethoxymethyl)phosphinic acid; Tocris Bioscience, Bristol, UK), and γ- Aminobutyric acid (Tocris Bioscience, Bristol, UK) were dissolved in distilled water. 1 mM concentration aliquots were stored at –20 °C or 4 °C and diluted with oxygenated aCSF to final concentration before bath application on slices.

## Data analysis

Demon Voltammetry and Analysis software was used to acquire and model FSCV data (*Yorgason et al., 2011*). Recording electrodes were calibrated by recording electrical current responses (in nA) to a known concentration of dopamine (3 mM) using a flow-injection system. This was used to convert electrical current to dopamine concentration. Michaelis-Menten kinetics were used to determine the maximal rate of dopamine uptake (*Vmax*) (*Ferris et al., 2013*).

## Statistical analysis

Baseline dopamine release to single-pulse stimulations and *Vmax* (raw numbers and percent change) were compared by students t-test. Baseline dopamine release to multi-pulse stimulations and

paired-pulse ratio were compared by two-way mixed-factor ANOVA. Similarly, baseline GABA and post α-Ctx GABA concentrations between adolescents and adults were compared by two-way mixed-factor ANOVA. Dopamine release following drug application (including percent change, normalized to 1 pulse pre-drug baseline, and paired-pulse ratio) were compared by three-factor generalized linear mixed model analysis. In the case of significant interactions, Bonferroni or Tukey post-hoc comparisons were used. Graph Pad Prism (version 8, La Jolla, CA) or SPSS (version 24, International Business Machine Corporation, Armonk, NY) were used to statistically analyze data sets (with $\alpha$ δ 0.05) and compose graphs. Values >2 standard deviations above or below the mean were considered outliers and excluded. This resulted in two adult rats and one adolescent rat being removed from data shown in *Figure 1A* because dopamine release was an outlier in all three cases. Data are presented as mean ± SEM for group data across multiple variables or mean with individual data points for bar graphs. For some experiments, testing multiple drugs on slice is a within-subject factor. However, each individual data point is representative of a single brain slice from one rat per group.

## Acknowledgements

This work was supported by the National Institutes of Health grants R01 DA052460 (MJF), R00 DA031791 (MJF), P50 DA006634 (MJF), P50 AA026117 (MJF), K12 GM102773 (EGP), F32 AA028162 (EGP), F31 DA058517 (MCI) and the Peter McManus Charitable Trust.

## Additional information

### Funding

| Funder | Grant reference number | Author |
| --- | --- | --- |
| National Institute on Drug Abuse | DA031791 | Mark J Ferris |
| National Institute on Drug Abuse | DA006634 | Mark J Ferris |
| National Institute on Alcohol Abuse and Alcoholism | AA026117 | Mark J Ferris |
| National Institute on Alcohol Abuse and Alcoholism | AA028162 | Elizabeth G Pitts |
| National Institute of General Medical Sciences | GM102773 | Elizabeth G Pitts |
| Peter McManus Charitable Trust | | Mark J Ferris |
| National Institute on Drug Abuse | DA058517 | Melody C Iacino |
| National Institute on Drug Abuse | DA052460 | Mark J Ferris |

The funders had no role in study design, data collection and interpretation, or the decision to submit the work for publication.

### Author contributions

Melody C Iacino, Conceptualization, Data curation, Formal analysis, Investigation, Methodology, Writing – original draft, Project administration, Writing – review and editing; Taylor A Stowe, Formal analysis, Investigation, Methodology, Writing – original draft, Writing – review and editing; Elizabeth G Pitts, Conceptualization, Formal analysis, Supervision, Investigation, Visualization, Methodology, Writing – original draft, Project administration, Writing – review and editing; Lacey L Sexton, Formal analysis, Investigation, Methodology, Project administration, Writing – review and editing; Shannon L Macauley, Conceptualization, Methodology, Writing – review and editing; Mark J Ferris,

Conceptualization, Resources, Formal analysis, Supervision, Funding acquisition, Visualization, Methodology, Writing – original draft, Project administration, Writing – review and editing

## Author ORCIDs
Melody C Iacino (iD) http://orcid.org/0000-0002-4807-2308
Mark J Ferris (iD) https://orcid.org/0000-0002-0127-7955

## Ethics
All animals were maintained according to the National Institutes of Health guidelines in Association for Assessment and Accreditation of Laboratory Animal Care accredited facilities (Accreditation #: 00008; PHS Assurance #: D16-00248 (A3391-01)). All experimental protocols were approved by the Institutional Animal Care and Use Committee at Wake Forest School of Medicine (Protocol Approval #: A21-143).

## Decision letter and Author response
Decision letter https://doi.org/10.7554/eLife.62999.sa1
Author response https://doi.org/10.7554/eLife.62999.sa2

## Additional files

### Supplementary files
• Transparent reporting form

### Data availability
All data generated or analyzed for this study are available at Harvard dataverse (https://dataverse.harvard.edu/) under the title eLife_2024_Iacino_Ferris, or through the following link: https://doi.org/10.7910/DVN/8D2FJ0.

The following dataset was generated:

| Author(s) | Year | Dataset title | Dataset URL | Database and Identifier |
|-----------|------|---------------|-------------|------------------------|
| Ferris MJ | 2024 | eLife_2024_Iacino_Ferris | https://dataverse.harvard.edu/dataset.xhtml?persistentId=doi:10.7910/DVN/8D2FJ0 | Harvard Dataverse, 10.7910/DVN/8D2FJ0 |

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
