## [Editor Report]

This is an important demonstration of how α-6 nicotinic ACh receptors regulate dopamine release via GABAergic mechanisms using ex vivo voltammetry recordings coupled with pharmacological manipulations. There is solid evidence provided to show that GABA tends to suppress dopamine release in adolescents but does not affect dopamine release in adults, a finding that is novel and interesting. Together the data will be of broad interest for further understanding multi-synaptic regulation of dopamine release.

---

## [Decision Letter]

**Decision letter after peer review:**

Thank you for submitting your article "A unique multi-synaptic mechanism regulates dopamine release in the nucleus accumbens during adolescence" for consideration by *eLife*. Your article has been reviewed by 3 peer reviewers, and the evaluation has been overseen by a Reviewing Editor, Shelly Flagel, and Senior Editor, Michael Taffe. The individuals involved in review of your submission have agreed to reveal their identity: Donita Robinson, Matthew Wanat, and Jamie McCutcheon.

The reviewers have discussed the reviews with one another and the Reviewing Editor has drafted this decision to help you prepare a revised submission.

The editors have judged that your manuscript is of interest, but, given the comments outlined below, additional experiments may be required before it is published. Thus, we would like to draw your attention to changes in our revision policy that we have made in response to COVID-19 (https://elifesciences.org/articles/57162). First, because many researchers have temporarily lost access to the labs, we will give authors as much time as they need to submit revised manuscripts. We are also offering, if you choose, to post the manuscript to bioRxiv (if it is not already there) along with this decision letter and a formal designation that the manuscript is "in revision at *eLife*". Please let us know if you would like to pursue this option. (If your work is more suitable for medRxiv, you will need to post the preprint yourself, as the mechanisms for us to do so are still in development.)

Summary:

The findings reported in this manuscript suggest that dopamine release is regulated by a unique multisynaptic mechanism that is evident during adolescence, but not in adult male rats. The data presented suggests that this mechanism involves acetycholine acting at α-6-containing nicotinic acetycholine receptors, which mediate inhibition of dopamine via GABA release. The Reviewers agree that the results are suggestive of a novel synaptic mechanism that could potentially serve as a biomarker for substance abuse vulnerability. However, there are a number of concerns raised, as indicated below. Primary concerns surround the use of a single approach to identify the proposed mechanism, inconsistencies in data presentation and analyses, and the generalizability of the findings.

Essential revisions:

1. I agree that the ex vivo method provides strong advantages to a mechanistic pharmacological study as the one presented here. However, it is also complicated by the use of electrical stimulation to evoke dopamine release. For example, acetylcholine and GABA will be released along with dopamine. Please specifically address this in the discussion. Related to this, please provide discussion to bridge the ex vivo measurements to in vivo measurements from the literature: do the findings reported here predict the in vivo observations, or are they at least consistent?

– Evoked dopamine in anesthetized rats: Cocaine increases stimulated dopamine release more in periadolescent than adult rats. Walker QD, Kuhn CM. Neurotoxicol Teratol. 2008 Sep-Oct;30(5):412-8. doi: 10.1016/j.ntt.2008.04.002. Epub 2008 Apr 22. PMID: 18508233.

– Spontaneous dopamine in awake rats: Fast dopamine release events in the nucleus accumbens of early adolescent rats. Robinson DL, Zitzman DL, Smith KJ, Spear LP. Neuroscience. 2011 Mar 10;176:296-307. doi: 10.1016/j.neuroscience.2010.12.016. Epub 2010 Dec 20. PMID: 21182904

2. Walker and Kuhn (2008) also reported reduced evoked dopamine release in male adolescent rats compared to adults in the caudate and nucleus accumbens. Analysis of the evoked signal revealed both reduced dopamine release per impulse as well as slower Vmax; in addition, Km in the caudate was lower in adolescent rats compared to adults, indicating higher affinity of the dopamine transporter for dopamine. If the authors can estimate Km from the present data? If so, adding this information will help to fill out the picture of age-related differences in dopamine regulation.

3. The authors' previous study reported marked sex differences in the comparison of adolescent and adult evoked dopamine release. Namely, while evoked, ex vivo dopamine release was lower in adolescent males than adult males, the opposite was observed in females. Only males were tested in the present study. Thus, the nACh/GABA mechanisms described here is not likely to generalize to females. This topic should be discussed.

4. Please provide the details on the voltage sweeps and the sampling frequency for the FSCV recordings. Most studies using FSCV perform voltage sweeps every 0.1s. This is important to state because it is not clear that this approach has the temporal resolution to examine PPR with the ISIs used in this study (0.01-0.2s). This issue is present in Figure 1E and Figure 4B.

5. Representative color plots and/or traces of FSCV recordings need to be included.

6. The normalization to the single pulse pre-drug dopamine response is appropriate for experiments involving only one pharmacological manipulation (e.g. Figure 2B). However, this pre-drug normalization is a problematic with experiments involving sequential pharmacological manipulations. For example, Figure 2D examines the effect of DHBE when a-Ctx is already on board. The data in this figure are normalized to the pre-drug dopamine response. Since the goal is to examine the effect of DHBE, a more appropriate normalization should be to the single pulse dopamine response with a-Ctx on board.

7. It is not clear why some of the data are normalized to the single pulse dopamine response (Figures2-3) and some of the data are presented as a % relative to a baseline treatment for data point (Figure 4). In general, I find the analysis in Figure 4 to be a bit more confusing because it is obscuring the increase in dopamine release with higher frequency stimulations.

8. Relating to the previous point, it appears that the decrease in DA release in adults following a-Ctx treatment is not blocked by GABA receptor antagonists (Figure 4E). However, this potentially interesting finding is not statistically examined since the data is presented as a % relative to the pre-Ctx condition.

9. Does GABA application differentially decrease dopamine release between adolescents and adults? This cannot be determined with how the data is presented in Figure 4F. This is an important point to address based on the model that is proposed. The observed results in juveniles could be explained by (i) greater expression of a6 receptors on GABA interneurons during adolescence or (ii) a greater sensitivity of dopamine neurons to GABA inhibition during adolescence (with no age-related changes in a6 receptor expression).

10. It seems that Figure 4A would be better placed in Figure 2 along with the representative examples (see comment #2). The same goes for Figure 4B, though I have my concerns if this analysis is appropriate (see comment #1).

11. Overlaying up to 8 traces with very subtle differences in color/symbol shape/line pattern makes it very challenging to visualize the data.

12. The schematics are a source of confusion. For example, there is no possible mechanism provided in Figure 2A by which a nicotinic receptor antagonist could increase dopamine release, as is shown in Figure 2C. I would suggest including a summary schematic at the end of the manuscript that illustrates the proposed differences between juveniles and adults.

13. My main concern is that the primary conclusion – i.e. that in early adolescence dopamine release is modulated by alpha6 nAChRs on GABA neurons in nucleus accumbens and that this mechanism does not exist in adulthood – is demonstrated through a single experimental paradigm. The data seem solid but it would give more confidence if this had been confirmed through an additional technique or in a less artificial situation. For example, biochemical, anatomical, or in vivo data to support their model or even data showing the consequences of this synaptic mechanism for the behavioral alterations associated with adolescence. Whether this level of additional depth is essential for publication though is a matter for discussion with the editor.

14. I have some concerns with the numbers of subjects used, the unit of statistic, and the possible outlier removal. For Figure 1 there are different numbers in each graph – if an outlier is removed from one measurement, I would expect them to be removed from the study or else it becomes difficult to interpret different measurements.

15. The use of a single concentration of the drugs leaves open the possibility that there is a shift in the concentration response curve rather than a qualitative difference in the underlying circuitry.

16. There are a number of places where the data are normalized to a value but the authors could be clearer about how this value might have been changed by the manipulation. For example, data in Figure 2B-D are normalized to 1 pulse but I found it a little difficult to ascertain how the 1 pulse data were affected by the manipulation. This is partly due to lack of description in the text and also due to the plotting style where it is difficult to distinguish symbols.

17. The use of paired-pulse ratio should be described more fully. In voltammetry, the method of calculating PPR and the conclusions that can be drawn this paradigm differ from traditional PPR as used in patch-clamp electrophysiology. It would be good if the authors are more explicit with this. In addition, they could discuss how the lack of age-dependent change in PPR (caused by α-Ctx) is not in conflict with the effects of a-Ctx using the other stimulation parameters.

18. More information should be stated on whether rats were bred in-house or ordered in. And if ordered in, how long was given for habituation. Potential stress of travel could be a factor.

[Editors' note: further revisions were suggested prior to acceptance, as described below.]

Thank you for resubmitting your work entitled "A unique multi-synaptic mechanism involving acetylcholine and GABA regulates dopamine release in the nucleus accumbens through early adolescence" for further consideration by *eLife*. Your revised article has been evaluated by me as both Senior Editor and Reviewing Editor, in collaboration with peer reviewers.

The manuscript has been improved significantly but there are some remaining issues that need to be addressed, as outlined below:

The authors are encouraged to address all of the specific points raised by the reviewers, but I wish to highlight several issues that were the focus of our discussions.

1) Consideration of the role of sex as a biological variable is essential, and it is appreciated that the manuscript refers to the prior work in the Introduction. This sex difference must be cued to the reader by adding the specification of male rats to the title and abstract. It would likewise be helpful to mention this briefly in the Discussion.

2) Data presentation: It is essential to use a consistent y-axis and data transformation for similar experiments to guide the reader. In addition, bar graphs should include the individual data points, consistent with current field expectations.

3) The inconsistencies of some of the data with the model advanced (as described by Reviewer #2) requires additional navigation, with options of either providing additional support or modifications of the model. At a minimum, there should be convincing speculative explanation for the apparent lack of evidence.

*Reviewer #1 (Recommendations for the authors):*

I appreciate the changes the authors have made to address reviewer concerns including adding data, although it was difficult to find some of the changes as the manuscript was not marked (e.g. changes in a different color font) and the responses did not indicate page or line numbers. The microdialysis data were helpful. Some additional items remain that I think will improve the readability, accuracy and impact of the manuscript.

1. Thank you for changing the title to "through early adolescence." Please do the same in the abstract, perhaps "… a multisynaptic mechanism potentially unique to the period of development that includes early adolescence…"

2. In results headers and figure captions, please indicate that these data are from males. For example, on page 7 line 120, "Adolescent male rats have decreased dopamine release in the NAc core". Only figures 1 and 3 indicate the data are in males, and the word "males" does not show up in the results. The schematics also should indicate males, as your lab has evidence that this is not what happens in females, or at least not on the same developmental track. Being super clear in this paper is not only accurate, but it will make more sense when data on females (presumably from your lab) are published that do not completely align.

3. Page 11, line 243: the findings in Robinson et al. 2011 are misrepresented. There was no difference in baseline dopamine transients in male adolescent rats (P 29-30) and adults (P 71-72), per abstract. (The Walker and Kuhn findings appear accurately represented.)

*Reviewer #2 (Recommendations for the authors):*

There are a couple of very interesting observations: (1) the effect of CTX on electrically-evoked dopamine release in adolescents and (2) the diverging effect of GABA on dopamine release between adolescents and adults – specifically the increase in dopamine release in adults. However, compelling evidence is not provided to support the model and it is not clear if these observations are necessarily linked. Specific comments are noted below, which are largely focused on the new additions to the manuscripts.

1) It is commendable that the authors performed the microdialysis experiments examining GABA levels in the NAc, which illustrated that local a-CTX injections reduced GABA levels only in adolescents. It is also appreciated that the authors now included the experiments examining the impact of GABA application in both adults and adolescents.

2) While the microdialysis data is consistent with the proposed model, there are other sets of data in this manuscript that contradict the model. For the model to work, there is an assumption that there is some endogenous level of ACh activation on a6 nicotinic receptors on GABA neurons, which is functioning to provide an inhibitory input onto dopamine neurons in brain slices from adolescents. Blocking a6 nicotinic receptors removes this inhibitory influence, which accounts for the increase in electrically-evoked dopamine release and lower GABA levels as assessed with microdialysis. A corollary of this model is that there must be endogenous GABA input onto dopamine terminals in slices. However, the absence of any effect of BIC or CGP on electrically-evoked dopamine release (Figure 4B) argues that there is not an appreciable GABA tone acting on dopamine terminals in slices. This discrepancy is not accounted for and largely disproves the proposed model.

3) The data showing that GABA tends to suppress dopamine release in adolescents but increases dopamine release in adults is quite novel and interesting. However, it is unclear how (and if) this is linked to the noted phenomenon regarding the a-CTX experiments.

4) Voltammetry traces that are presented are not reflecting the average trend. This is notable in Figures3-4 with the MLA effect relative to baseline (should be smaller in both groups but that is not the case with the adults). The same concerns are also evident with the a-CTX experiment where the mid adolescent shows a higher DA with the 1p stimulation.

5) Issues still remain regarding the normalization used between the figures. The y-axis on the primary subpanels in Figures 2-3 refer to 'Dopamine release normalized to 1 Pulse' whereas the y-axis in Figure 4 refer to 'Dopamine release (%Baseline)'. Are these the same metrics, except that Figure 4 is converting the values into percentages? The rationale for doing so is not evident and is a source of confusion.

6) The introduction now discusses sex differences and provides a rationale for only focusing on males. While one can appreciate the challenges that Covid had on labs, it has been quite some time since the NIH introduced its SABV policy. The current policy of many other journals is that single-sex studies must state the sex studied in the title and abstract.

*Reviewer #3 (Recommendations for the authors):*

The authors have made significant changes to the manuscript that are very much appreciated.

I have just a few remaining comments that they may or may not wish to act on.

p.10, l.11-13 – In the new results from the in vivo dialysis study, the authors state "Interestingly, we found that infusion of α-Ctx into the NAc did not modulate GABA tone differently in adults and adolescents" (main effect of drug: F1,27=0.69, p=0.4109)" but do not provide the interaction term from this ANOVA, which is an important statistic.

In the Discussion, they also state, "Moreover, infusing α-Ctx directly into the NAc reduced GABA tone in adolescent, but not adult, rats.". I am not sure whether the data support this claim.

"400 mm-thick coronal slices" should be μm

"recording microelectrode was placed 100-150 mM" should be mm

"a triangular waveform (-0.4 to +1.2 to -0.4 V vs Ag/Agcl, 400 Vs)" should be V/s

"This resulted in 2 adult rats and 1 adolescent rat being removed from data shown in igure 1A" should be Figure

[Editors' note: further revisions were suggested prior to acceptance, as described below.]

Thank you for resubmitting your work entitled "A unique multi-synaptic mechanism involving acetylcholine and GABA regulates dopamine release in the nucleus accumbens through early adolescence in male rats" for further consideration by *eLife*. Your revised article has been evaluated by Michael Taffe as both Senior Editor and Reviewing Editor in consultation with several peer reviewers.

The manuscript has been improved but there are some remaining issues that need to be addressed, as outlined below:

As you can see below, one reviewer had some concerns about the microdialysis data. In our pre-decision discussions the other reviewers and I concurred with these points and endorsed the need for some additional modifications.

*Reviewer #3 (Recommendations for the authors):*

The manuscript has been improved by the latest round of revisions. My final point relates to presentation of the microdialysis data that I feel still needs revision due to discrepancies between the Results, Discussion and the Figure Caption.

The Results provide statistics showing the interaction between Age x Drug allowing the conclusion that α-Ctx acts differently at different ages. However, no post hoc values are provided here though, which precludes specific statements regarding increases or decreases within or between different groups. The Discussion states various differences (higher GABA tone in adolescents, drug effect to decrease in adolescents and increase in adults), which should be backed up by these group comparisons. The figure caption seems in conflict with these sections as it states "Interestingly, infusing α-Ctx into the NAc did not differentially affect adult or adolescent GABA levels, but revealed that adolescents have higher GABA tone as compared to adult." The figure itself has a star to indicate significance but it is unclear where this comes from or which group the comparison is supposed to apply to.

---

## [Author Response]

Essential revisions:1. I agree that the ex vivo method provides strong advantages to a mechanistic pharmacological study as the one presented here. However, it is also complicated by the use of electrical stimulation to evoke dopamine release. For example, acetylcholine and GABA will be released along with dopamine. Please specifically address this in the discussion. Related to this, please provide discussion to bridge the ex vivo measurements to in vivo measurements from the literature: do the findings reported here predict the in vivo observations, or are they at least consistent?– Evoked dopamine in anesthetized rats: Cocaine increases stimulated dopamine release more in periadolescent than adult rats. Walker QD, Kuhn CM. Neurotoxicol Teratol. 2008 Sep-Oct;30(5):412-8. doi: 10.1016/j.ntt.2008.04.002. Epub 2008 Apr 22. PMID: 18508233.– Spontaneous dopamine in awake rats: Fast dopamine release events in the nucleus accumbens of early adolescent rats. Robinson DL, Zitzman DL, Smith KJ, Spear LP. Neuroscience. 2011 Mar 10;176:296-307. doi: 10.1016/j.neuroscience.2010.12.016. Epub 2010 Dec 20. PMID: 21182904

This is a good point. We now discuss the differences/similaries between in vivo and ex vivo voltammetry studies and attempt to bridge our study with adolescent in vivo studies in introduction and discussion. (including citations of Walker and Kuhn, 2008 and Robinson et al., 2011)

2. Walker and Kuhn (2008) also reported reduced evoked dopamine release in male adolescent rats compared to adults in the caudate and nucleus accumbens. Analysis of the evoked signal revealed both reduced dopamine release per impulse as well as slower Vmax; in addition, Km in the caudate was lower in adolescent rats compared to adults, indicating higher affinity of the dopamine transporter for dopamine. If the authors can estimate Km from the present data? If so, adding this information will help to fill out the picture of age-related differences in dopamine regulation.

This is an interesting point. After much reanalysis of data, we elected not to include variability in Km data for this paper because we are not fully confident that we can model Km in our ex vivo preparation. There have been reports of Km in prior publications by Ferris that have ranged from 160 nM to 200 nM and is consistent with reports from seminal papers, but in those studies the slices and testing conditions were painstakingly optimized for detection of Km at the bottom portion of the descending limb of dopamine traces. While we used good signals that can model release and Vmax for the current study, we really did not optimize for Km assessment. We included this point in the discussion about how we did not optimize for Km but that Walker and Kuhn have shown this effect.

3. The authors' previous study reported marked sex differences in the comparison of adolescent and adult evoked dopamine release. Namely, while evoked, ex vivo dopamine release was lower in adolescent males than adult males, the opposite was observed in females. Only males were tested in the present study. Thus, the nACh/GABA mechanisms described here is not likely to generalize to females. This topic should be discussed.

We now discuss sex differences in the introduction to address this limitation.

4. Please provide the details on the voltage sweeps and the sampling frequency for the FSCV recordings. Most studies using FSCV perform voltage sweeps every 0.1s. This is important to state because it is not clear that this approach has the temporal resolution to examine PPR with the ISIs used in this study (0.01-0.2s). This issue is present in Figure 1E and Figure 4B.

We included more information on the voltage sweeps and sampling frequency in the methodology. Note that we do sample every 0.1s so the reviewer is correct that we are not able to resolve DA magnitude for both pulses in the two-pulse condition. Our paired pulse follows current approaches in the literature from Alvarez and Cragg groups among others. It is the ratio of the single peak height (not separable) of the 2 pulses (summed), minus the 1 pulse peak height, over the peak height that is evoked from 1 pulse (i.e., P2-P1/P1). This is consistent with the literature that uses this approach from other laboratories, and has been explored mechanistically by other labs as well (Condon et a., 2019; Cragg et al., 2003). We now clarify this in the methodology.

5. Representative color plots and/or traces of FSCV recordings need to be included.

Representative traces of FSCV recordings have been added to each figure.

6. The normalization to the single pulse pre-drug dopamine response is appropriate for experiments involving only one pharmacological manipulation (e.g. Figure 2B). However, this pre-drug normalization is a problematic with experiments involving sequential pharmacological manipulations. For example, Figure 2D examines the effect of DHBE when a-Ctx is already on board. The data in this figure are normalized to the pre-drug dopamine response. Since the goal is to examine the effect of DHBE, a more appropriate normalization should be to the single pulse dopamine response with a-Ctx on board.

This is now revised in the manuscript according to the reviewer’s comment. We were not clear in the first draft on the analysis and our approach which we think lead to confusion. We retained the original graph with all drugs relative to no drug baseline in Figure 2D and now include the reviewer’s suggestion of DHBE (as drug #2) relative to conotoxin (as drug #1) in Figure 2E.

7. It is not clear why some of the data are normalized to the single pulse dopamine response (Figures2-3) and some of the data are presented as a % relative to a baseline treatment for data point (Figure 4). In general, I find the analysis in Figure 4 to be a bit more confusing because it is obscuring the increase in dopamine release with higher frequency stimulations.

We were not clear in the first version of the manuscript as to what the percent baseline data were. Figures 2 and 4 now show a percent of baseline that is relative to the drug free condition for all drugs, as well as drug #2 in relation to drug #1 given comments from other reviewers about these same data. (see #6 response above).

8. Relating to the previous point, it appears that the decrease in DA release in adults following a-Ctx treatment is not blocked by GABA receptor antagonists (Figure 4E). However, this potentially interesting finding is not statistically examined since the data is presented as a % relative to the pre-Ctx condition.

Similar to points 6 and 7 on normalization, we are hoping that we have clarified our miscommunication, and we have added data. DA release in adult rats following a-ctx is not altered by GABA receptor antagonists. We know that α-ctx modulates DA release independent of GABA release via direct interactions with DA terminals, not only this result, but many prior studies.

9. Does GABA application differentially decrease dopamine release between adolescents and adults? This cannot be determined with how the data is presented in Figure 4F. This is an important point to address based on the model that is proposed. The observed results in juveniles could be explained by (i) greater expression of a6 receptors on GABA interneurons during adolescence or (ii) a greater sensitivity of dopamine neurons to GABA inhibition during adolescence (with no age-related changes in a6 receptor expression).

Thank you for this point. GABA application to slices does interact with age to differentially affect dopamine release, which is now shown in Figure 4, included in results, and discussed in the manuscript.

10. It seems that Figure 4A would be better placed in Figure 2 along with the representative examples (see comment #2). The same goes for Figure 4B, though I have my concerns if this analysis is appropriate (see comment #1).

This is a good idea. We rearranged figure 2 to include old Figure 4A and 4B plus relevant traces (see #11).

11. Overlaying up to 8 traces with very subtle differences in color/symbol shape/line pattern makes it very challenging to visualize the data.

This has been corrected. Traces have been separated onto new graphs where appropriate. We have also adjusted the y-axes to ensure the full range of data effects are able to be easily visualized.

12. The schematics are a source of confusion. For example, there is no possible mechanism provided in Figure 2A by which a nicotinic receptor antagonist could increase dopamine release, as is shown in Figure 2C. I would suggest including a summary schematic at the end of the manuscript that illustrates the proposed differences between juveniles and adults.

Thank you bringing this to our attention. We elected to include an additional schematic at the end of the manuscript to illustrate the proposed age-related differences (see Figure 5). Our schematic describes the differential GABAmediated circuits in adults and adolescents by presenting a side-by-side comparison of both baseline and post α-Ctx proposed mechanisms in the NAc core.

13. My main concern is that the primary conclusion – i.e. that in early adolescence dopamine release is modulated by alpha6 nAChRs on GABA neurons in nucleus accumbens and that this mechanism does not exist in adulthood – is demonstrated through a single experimental paradigm. The data seem solid but it would give more confidence if this had been confirmed through an additional technique or in a less artificial situation. For example, biochemical, anatomical, or in vivo data to support their model or even data showing the consequences of this synaptic mechanism for the behavioral alterations associated with adolescence. Whether this level of additional depth is essential for publication though is a matter for discussion with the editor.

We now include in vivo measurements of GABA both pre- and post-α-ctx infusion using microdialysis in the nucleus accumbens in awake, freely-moving rats. The results of these data are presented in Figure 4H as well as in the proposed mechanism for Figure 5. Our in vivo studies provide parallel support to our ex vivo studies. Additional thoughts on these results have also been highlighted in the discussion.

14. I have some concerns with the numbers of subjects used, the unit of statistic, and the possible outlier removal. For Figure 1 there are different numbers in each graph – if an outlier is removed from one measurement, I would expect them to be removed from the study or else it becomes difficult to interpret different measurements.

We agree on this point. We were unclear about some of the methodology in the original submission, but have since clarified the unit of statistic as well as the number of subjects used in each study throughout the paper (see minor comment #1, #6). Note that only 3 outliers were removed, and that different numbers will be used across drugs / figures because we have multiple rigs that can run many slices / drugs from the same animal. However, sometimes there are not enough rigs or slices to run all drugs, so another animal will be run to compensate and over time, different numbers are run depending on prioritization of drug or to achieve sufficient power for varying effect sizes.

15. The use of a single concentration of the drugs leaves open the possibility that there is a shift in the concentration response curve rather than a qualitative difference in the underlying circuitry.

This is a good point. We admit to being agnostic in this broad screening of compounds to shifts in potency vs efficacy, and now acknowledge directly in the manuscript this limitation, namely that we cannot assess these differences without the concentration response curve. However, we do believe that shifts in either (or both) potency and efficacy would be a result of differences in the biology of the circuit. For example, we know that receptor number might influence potency and therefore shift the concentration of neurotransmitter needed to activate a postsynaptic cell. Admittedly, we can’t test whether such things can be overcome (efficacy) with enough neurotransmitter. Future studies, paired with histochemical assessment of receptor density, for example, will be needed. This is all now addressed in the manuscript.

16. There are a number of places where the data are normalized to a value but the authors could be clearer about how this value might have been changed by the manipulation. For example, data in Figure 2B-D are normalized to 1 pulse but I found it a little difficult to ascertain how the 1 pulse data were affected by the manipulation. This is partly due to lack of description in the text and also due to the plotting style where it is difficult to distinguish symbols.

This is now made clearer in the revised version. What each data is normalized to is now made more clear. Note that normalizing to drug free conditions is used in other literature and our goal was to make similar to similar work (e.g., Rice and Cragg, 2004, *Nature Neuroscience)* similar to how other classic data are presents, like microdialysis data, whereby each time-point (in this case frequency), is normalized to the drug free condition.

17. The use of paired-pulse ratio should be described more fully. In voltammetry, the method of calculating PPR and the conclusions that can be drawn this paradigm differ from traditional PPR as used in patch-clamp electrophysiology. It would be good if the authors are more explicit with this. In addition, they could discuss how the lack of age-dependent change in PPR (caused by α-Ctx) is not in conflict with the effects of a-Ctx using the other stimulation parameters.

This is a good point. We have clarified our definition of PPR in the methodology (see #4). As per reviewer request, we have also highlighted this point in the discussion.

18. More information should be stated on whether rats were bred in-house or ordered in. And if ordered in, how long was given for habituation. Potential stress of travel could be a factor.

This has been clarified in the methodology. Rats were ordered from a vendor and allowed one full week in their home cage / chamber for habituation prior to experiments. This was the case for both adolescent and adult rats.

[Editors’ note: what follows is the authors’ response to the second round of review.]

The manuscript has been improved significantly but there are some remaining issues that need to be addressed, as outlined below:The authors are encouraged to address all of the specific points raised by the reviewers, but I wish to highlight several issues that were the focus of our discussions.1) Consideration of the role of sex as a biological variable is essential, and it is appreciated that the manuscript refers to the prior work in the Introduction. This sex difference must be cued to the reader by adding the specification of male rats to the title and abstract. It would likewise be helpful to mention this briefly in the Discussion.

We now include the wording of “in male rats” in the title, in every figure legend title, and in every results header. We kept our inclusion of the word “male rats” in the abstract (…in the nucleus accumbens (NAc) core of adolescent (P28-35) and adult (P70-90) male rats) and added an additional mention of “male rat.” We also added several mentions of male rats throughout the manuscript in every section. Per your comment, we have added some brief discussion on sex differences to match that of the introduction.

2) Data presentation: It is essential to use a consistent y-axis and data transformation for similar experiments to guide the reader. In addition, bar graphs should include the individual data points, consistent with current field expectations.

Thank you for bringing this to our attention. We have adjusted the y-axis in figures 2, 3, and 4 to maintain consistency. We now include more detail on analysis in each of the figure legends as well as detailed explanations throughout the results to provide clarity. Graph titles have also been adjusted in figures 2-4 to reflect those changes. Additionally, we have kept the individual data points shown in figures 1-3 and have added the individual data points that were missing from the bar graph in figure 4.

3) The inconsistencies of some of the data with the model advanced (as described by Reviewer #2) requires additional navigation, with options of either providing additional support or modifications of the model. At a minimum, there should be convincing speculative explanation for the apparent lack of evidence.

See response to reviewer #2, comment #1

Reviewer #1 (Recommendations for the authors):I appreciate the changes the authors have made to address reviewer concerns including adding data, although it was difficult to find some of the changes as the manuscript was not marked (e.g. changes in a different color font) and the responses did not indicate page or line numbers. The microdialysis data were helpful. Some additional items remain that I think will improve the readability, accuracy and impact of the manuscript.1. Thank you for changing the title to "through early adolescence." Please do the same in the abstract, perhaps "… a multisynaptic mechanism potentially unique to the period of development that includes early adolescence…"

We have now revised this sentence in the abstract.

2. In results headers and figure captions, please indicate that these data are from males. For example, on page 7 line 120, "Adolescent male rats have decreased dopamine release in the NAc core". Only figures 1 and 3 indicate the data are in males, and the word "males" does not show up in the results. The schematics also should indicate males, as your lab has evidence that this is not what happens in females, or at least not on the same developmental track. Being super clear in this paper is not only accurate, but it will make more sense when data on females (presumably from your lab) are published that do not completely align.

Thank you for this comment. We now include the mention of “male rats” in the title, every subheading, and throughout the Results section *(see response to comment #1)*.

3. Page 11, line 243: the findings in Robinson et al. 2011 are misrepresented. There was no difference in baseline dopamine transients in male adolescent rats (P 29-30) and adults (P 71-72), per abstract. (The Walker and Kuhn findings appear accurately represented.)

Thank you for this comment. We now include the mention of “male rats” in the title, every subheading, and throughout the Results section *(see response to comment #1)*.

Reviewer #2 (Recommendations for the authors):There are a couple of very interesting observations: (1) the effect of CTX on electrically-evoked dopamine release in adolescents and (2) the diverging effect of GABA on dopamine release between adolescents and adults – specifically the increase in dopamine release in adults. However, compelling evidence is not provided to support the model and it is not clear if these observations are necessarily linked. Specific comments are noted below, which are largely focused on the new additions to the manuscripts.1) It is commendable that the authors performed the microdialysis experiments examining GABA levels in the NAc, which illustrated that local a-CTX injections reduced GABA levels only in adolescents. It is also appreciated that the authors now included the experiments examining the impact of GABA application in both adults and adolescents.2) While the microdialysis data is consistent with the proposed model, there are other sets of data in this manuscript that contradict the model. For the model to work, there is an assumption that there is some endogenous level of ACh activation on a6 nicotinic receptors on GABA neurons, which is functioning to provide an inhibitory input onto dopamine neurons in brain slices from adolescents. Blocking a6 nicotinic receptors removes this inhibitory influence, which accounts for the increase in electrically-evoked dopamine release and lower GABA levels as assessed with microdialysis. A corollary of this model is that there must be endogenous GABA input onto dopamine terminals in slices. However, the absence of any effect of BIC or CGP on electrically-evoked dopamine release (Figure 4B) argues that there is not an appreciable GABA tone acting on dopamine terminals in slices. This discrepancy is not accounted for and largely disproves the proposed model.

We agree with this insightful comment. We now include additional discussion (third from last paragraph, blue font) on how the GABA antagonists could be blocking the effect of α-conotoxin in adolescent animals while having no effect on their own. First, a new paper was just published showing that GABA does in fact provide inhibitory tone on dopamine release and, when blocked with GABA antagonists, dopamine can be disinhibited in that paper (Patel et al., 2024). That study, however, used optogenetics to selectively activate dopamine varicosities, whereas our study uses electrical stimulation. We now cite this study in the discussion which directly supports our model as well as discuss the differences in general electrical stimulation compared to selective stimulation may preclude our ability to detect increases in dopamine tone from GABA antagonism.

3) The data showing that GABA tends to suppress dopamine release in adolescents but increases dopamine release in adults is quite novel and interesting. However, it is unclear how (and if) this is linked to the noted phenomenon regarding the a-CTX experiments.

We agree that there may not be a link between the GABA alone study (Figure 4G) and the α-contoxin effect in general, and now include this point in the discussion (Third from last paragraph). Briefly here, GABA was designed to be administered prior to α-conotoxin for the main purpose to study the alpha6 effect. It was subsequently included on the revision because we agreed with a prior review comment that this effect might be interesting, but we agree the effect alone is likely not linked to α-conotoxin effect. We are happy to remove Figure 4G and the new discussion if such an alternative revision clarifies the paper.

4) Voltammetry traces that are presented are not reflecting the average trend. This is notable in Figures3-4 with the MLA effect relative to baseline (should be smaller in both groups but that is not the case with the adults). The same concerns are also evident with the a-CTX experiment where the mid adolescent shows a higher DA with the 1p stimulation.

Thank you for noting this. Yes, we originally picked traces at random to be as unbiased as possible which sometimes selects a trace that is on the fringe of overlapping group distributions. We now swap out these for the representative traces that represent the mean of the group which we agree is more appropriate. Edits are not marked but are in Figures 2 and 3 (Figure 4 does not have representative traces).

5) Issues still remain regarding the normalization used between the figures. The y-axis on the primary subpanels in Figures 2-3 refer to 'Dopamine release normalized to 1 Pulse' whereas the y-axis in Figure 4 refer to 'Dopamine release (%Baseline)'. Are these the same metrics, except that Figure 4 is converting the values into percentages? The rationale for doing so is not evident and is a source of confusion.

We now change everything to % of baseline for consistency and clarify what the baseline is in each Results section and figure legend for Figures 2 – 4. We used baselines that are consistent with prior slice voltammetry literature that highlights and clarify effects. (See editor comment #2).

6) The introduction now discusses sex differences and provides a rationale for only focusing on males. While one can appreciate the challenges that Covid had on labs, it has been quite some time since the NIH introduced its SABV policy. The current policy of many other journals is that single-sex studies must state the sex studied in the title and abstract.

We now include the use of “male rats” in the title as well as additional mentions of “male rats” throughout the abstract and results. (see editor comment #1).

Reviewer #3 (Recommendations for the authors):The authors have made significant changes to the manuscript that are very much appreciated.I have just a few remaining comments that they may or may not wish to act on.p.10, l.11-13 – In the new results from the in vivo dialysis study, the authors state "Interestingly, we found that infusion of α-Ctx into the NAc did not modulate GABA tone differently in adults and adolescents" (main effect of drug: F1,27=0.69, p=0.4109)" but do not provide the interaction term from this ANOVA, which is an important statistic.

Thank you for your comment. We agree on this point. We have corrected this initial oversight by including the significant interaction in the results to support our claim in the discussion. “Drug X Age: F_1,26_=6.408, p=0.0178” (Figure 4H, Figure 5).

In the Discussion, they also state, "Moreover, infusing α-Ctx directly into the NAc reduced GABA tone in adolescent, but not adult, rats.". I am not sure whether the data support this claim.

We have revised this sentence in the discussion.

"400 mm-thick coronal slices" should be μm

Corrected.

"recording microelectrode was placed 100-150 mM" should be mm

Corrected.

"a triangular waveform (-0.4 to +1.2 to -0.4 V vs Ag/Agcl, 400 Vs)" should be V/s

Corrected.

"This resulted in 2 adult rats and 1 adolescent rat being removed from data shown in igure 1A" should be Figure

Corrected.

[Editors’ note: what follows is the authors’ response to the third round of review.]

The manuscript has been improved but there are some remaining issues that need to be addressed, as outlined below:As you can see below, one reviewer had some concerns about the microdialysis data. In our pre-decision discussions the other reviewers and I concurred with these points and endorsed the need for some additional modifications.Reviewer #3 (Recommendations for the authors):The manuscript has been improved by the latest round of revisions. My final point relates to presentation of the microdialysis data that I feel still needs revision due to discrepancies between the Results, Discussion and the Figure Caption.The Results provide statistics showing the interaction between Age x Drug allowing the conclusion that α-Ctx acts differently at different ages. However, no post hoc values are provided here though, which precludes specific statements regarding increases or decreases within or between different groups. The Discussion states various differences (higher GABA tone in adolescents, drug effect to decrease in adolescents and increase in adults), which should be backed up by these group comparisons. The figure caption seems in conflict with these sections as it states "Interestingly, infusing α-Ctx into the NAc did not differentially affect adult or adolescent GABA levels, but revealed that adolescents have higher GABA tone as compared to adult." The figure itself has a star to indicate significance but it is unclear where this comes from or which group the comparison is supposed to apply to.

Thank you for this comment. We revised the Results, Figure Caption, and Discussion to be more precise with our language regarding the interaction and include the Sidak post hoc comparison to indicate that adolescent baseline GABA is significantly different than adult baseline GABA. We also indicate in the figure caption that the asterisk is indicative of the Sidak posthoc test. Note that a few sentences of more elaborate speculation was removed in these areas to be more precise with noting the interaction and the baseline posthoc effect.